# Health services satisfaction and medical exclusion among migrant youths in Gauteng Province of South Africa: A cross-sectional analysis of the GCRO survey (2017–2018)

**Monica Ewomazino Akokuwebe**[1]*, **Godswill Nwabuisi Osuafor**[2], **Salmon Likoko**[3], **Erhabor Sunday Idemudia**[1]

1 Faculty of Humanities, North-West University, Mafikeng, South Africa, 2 Population Studies and Demography, Faculty of Humanities, North-West University, Mafikeng, South Africa, 3 Statistics South Africa, Pretoria, South Africa

* nwupostdoctoral2021@gmail.com

**Data Availability Statement:** The data underlying the results presented in the study are available from (https://www.gcro.ac.za).

## Abstract

### Background

Medical xenophobia of migrant (either in-migrants or immigrants) youths is an ongoing problem in contemporary South African society. Medical mistreatment by healthcare workers and social phobia from migrant youths have been attributed to major obstacles to healthcare utilization as well as health services satisfaction. This study aimed to determine the prevalence and factors contributing to health services satisfaction and medical exclusion among migrant youths in Gauteng province in South Africa.

### Methods

The Round 5 Gauteng City-Region Observatory (GCRO) Quality of Life (QoL) survey was conducted in 2017–2018, a nationally representative survey piloted every two years in South Africa, was utilized in this study. A 2-year cohort study of 24,889 respondents aged 18 to 29 and a baseline data consisted of 4,872 respondents, comprising non-migrants, in-migrants and immigrants, from where 2,162 in-migrants and immigrants were utilized as the sample size. The data was analysed using descriptive statistics, Chi-Square analysis and logistic regression.

### Results

A total of 2,162 migrants, comprising 35.4% in-migrants and 9.0% of immigrants, from the 4,872 respondents, were included in the analysis. The prevalence of medical exclusion of in-migrant and immigrant youths were 5.5% and 4.2%, and the majority of them reported the use of public health facilities (in-migrants – 84.3% vs. immigrants – 87.1%). At the bivariate level, demographic (age, sex, and population group), economic (employed and any income) and health-related (no medical aid and household member with mental health) factors were significantly associated with medical exclusion ($p \leq 0.05$). The adjusted odds ratio showed

**Funding:** The author(s) received no specific funding for this work.

**Competing interests:** The authors have declared that no competing interests exist.

that only female gender (AOR: 1.07, 95% CI: 0.678, 1.705), no medical aid cover (AOR: 1.23, 95% CI: 0.450, 3.362), and neither (AOR: 1.59, 95% CI: 0.606, 4.174) or dissatisfied (AOR: 4.29, 95% CI: 2.528, 7.270) were independent predictors of medical exclusion.

## Conclusion

Having no medical aid cover, being a female and dissatisfied, or neither satisfied nor dissatisfied with health services significantly increased the odds of medical exclusion among migrant youths. To increase healthcare utilization and ensuring adequate medical care of migrant youths, opting for medical aid insurance without increasing costs should be guaranteed. Therefore, there should be no consequences for lack of residence status or correct documentation papers when accessing healthcare services among migrant youths in South Africa.

## Introduction

Across the globe, including the African continent, migration is being influenced significantly by economic instability, poverty, armed conflict and unrest, as well as social, political, and technological transformations [1–3]. Critical conditions push people from their home countries to seek a better life elsewhere. As a result of this mass relocation of people, public resources will be challenged in the different countries to which they are relocating. This is both during their movement and upon their arrival. Regardless of gender, migrants are often victims of violence, infectious diseases, and malnutrition, which frequently accompany displaced populations and migrant movements [4, 5]. Migrants nevertheless face more risks to their safety and wellbeing than the non-migrant, due to the abrupt and dramatic onset of emergencies and related uncertainty connected with migrant status, despite the massive efforts made to meet their health requirements [6, 7]. However, the complex interactions between the migrants' status and their health indicate that these may have either positive or negative effects on their overall well-being [8–11]. Therefore, it is critical for migrants to have access to healthcare services in their host countries.

The International Organization for Migration (2020) survey report showed that the present-day African migrants are largely youths, since 60.0% of irregular African migrants are under 35 years; and 27.5% of migrants aged 15–29 years are hosted in several African countries [1, 5]. Migration of youths from low-income to high-income nations with a trend and patterns of stability in political, social and economic domains. Nearly 39.5 million Africans have migrated, of which 21 million did so within the continent and 18.5 million outside of it [3, 5]. In Africa, labour migration is largely intra-regional (80%), of which a majority are low-skilled labour migrants. In addition, of great importance in the region, is the alliance of South-South migration corridors to nearby markets in search of employment with better remuneration [11–13]. African migrants transiting to other countries are in high demand in economic sectors. These economic sectors are significant drivers for heavy flow of migration routes within or outside the African continent.

Similarly, South Africa has been the destination for most migrants from other African countries, with agreement ties with the United Nations High Commissioner for Refugees (UNHCR) to manage migrant-related issues [14]. Even though their contributions to the economic, sociocultural, and political development of their host nations are seldom publicly

acknowledged, migrants nevertheless experience social marginalization and poor health [15]. These challenges are multifaceted, ranging from racial and ethnic to generational differences. Primarily, the poor health outcomes of migrants can be construed from how migrants juggle multiple menial jobs to earn the income required to meet their upkeep and contributions to their households [16, 17]. However, responses from existing health policy do not extend to migrant patients, hence facing a myriad problems when trying to access public health services in South Africa. But the right to have access to health care services is a basic fundamental human right guaranteed by the Constitution of any nation [18]. In South Africa, the Human Rights Commission states Section 27 of the Constitution that provides that: "*everyone has the right to have access to healthcare services, including reproductive health care services, and no one may be refused emergency medical treatment*" [19].

The Statistics South Africa reports have stated an estimation of 2.9 million migrants presently residing in South Africa at mid-year 2020. In 2023, the net migration was 1.840 per 1000 persons with a decrease of 7.02% from 2022, while a net migration of 1.979 was reported per 1000 persons, with a decline of 6.56% from 2021 [7, 20, 21]. To some extent though not entirely, some of the immigrant populations with employment have created a positive impact on government's fiscal balance and their contributions to government's budgetary health owing to their frequent payment of higher taxes [22, 23]. Also, with the implementation of the National Health Insurance (NHI) scheme, public healthcare funding presently comes from government spending via taxation and point-of-care expenses from those utilizing these health care services [22, 24, 25]. Forced migrant populations accounting for 9% of the whole documented immigrant population, struggle to access public services, including healthcare, despite the fact that they are legally eligible to access these services [25, 26]. Several studies have reported individuals are being turned away from government health facilities owing to immigration status, nationality or language spoken [25–27]. For instance, migrant women in specific have experienced various encounters when struggling to gain access to antenatal care, including at the time of delivery. Some studies conducted in South Africa have mentioned that some maternal healthcare facilities have declined migrant women who are unable to pay for maternal services to take home their newborn infants [22, 25, 27]. Thus, lack of health services satisfaction and medical exclusion of migrants can result in a wide risk population health epidemic disease and therefore, refusing to grant a share of the South African migrant population access to preventative and curative healthcare services, may sabotage and thwart invested efforts to control infectious diseases such as cholera, including HIV and tuberculosis.

As earlier mentioned, the background to the study majorly focused on in-migrants (South African nationals) and immigrants (foreign nationals), why they migrate and the impact on their health and the need to access healthcare. Thus, in-migrant's access to healthcare is a particular concern given the centrality of poor access in perpetuating inequality and poverty among South African nationals [22–25]. The apartheid history has left a large racial disparities in access despite post-apartheid health policy to increase the number of health facilities, even in inaccessible rural areas. For it to be xenophobic, medical treatment must be unjustly denied on the basis of one's nationality or legal stay [26, 27]. Also, there are other grounds that medical care might by wrongly denied, as South African healthcare system is found to be in an advanced state of disrepair and many staff can be highly stressed in such environments exhibiting xenophobic behaviours or attitudes. Socioeconomic factors, transportation, disabilities, and stigmatization have also been cited in studies that pose as barriers limiting a majority of South Africa in-migrants from accessing necessary health care [22, 23].

Similarly, the proportion of poverty is increasing among rural, posing a key risk on death proportions. Studies have shown that South African nationals, including in-migrants also faces discrimination in accessing medical care and the issue is not related to migrants' use of the

healthcare system but rather how health financing budgets to cover health expenses for all citizens, including migrants have been unsuccessful over the years [26–28]. Notably, internal migrants moving between provinces, accounted for much more than cross-border migration in SADC, as medical expenses are never budgeted for in-migrants in their new destinations/ province of relocation. Budget and fiscal planning are often based on obsolete population data, as in-migrants are not considered when planning healthcare budget distribution and, updating population register is very important in planning for such basic services [23, 28]. However, health policies regarding in-migrants' access to healthcare are not consistent, creating unclear situations for health workers on who can be treated. These gaps have not been addressed as the South African department of health has published memorandums which may confuse medical staff about in-migrants and immigrants' rights and their accessibility to healthcare [20, 21].

Factors (such as health status, language barrier, and migration status) associated with migrants' exclusion from healthcare access have been cited in several studies conducted in South Africa [28–30]. First, pre-migration factors (trauma experience) and post-migration challenges (access to healthcare services) have been reported to majorly contribute negatively to migrants' health [9, 31]. Other factors that may cause medical exclusion of migrants include poor knowledge of migrant rights, low socio-economic status, cultural variety and interests, religious beliefs, language barriers, and poor understanding of the healthcare systems in the host countries of the migrants [32, 33]. Other studies have mentioned several barriers hindering migrants from accessing healthcare, such as cost, health system complexity, incompetent health workers, long queues and wait times, discriminatory attitudes or mannerisms of health workers towards migrants with chronic ailments (such as HIV, TB, diabetes etc.), and fear among undocumented immigrants [34, 35]. Moreover, quite a number of existing studies have showed that socio-demographic factors such as educational level, income, employment, age, health insurance coverage, and surety (parent, guardian, or guarantor) have been mentioned as important social determinants of health [8, 28–30]. Thus, socioeconomic determinants of health play a key role in the decline of migrants' health after a period in their host nations [33], and this is particularly true for African migrants [3].

Also, other studies have mentioned medical xenophobia as one of the major barriers that migrants are faced with and this has been a hindrance to healthcare accessibility in South Africa [32–35]. Medical xenophobia has been labelled as having the forms of negative attitudes, unsupportive practices, denied access to any form of medical treatment or care, stigmatization, structural violence, stereotyping, and blaming migrants for their destitution [23, 27]. Also, there is a misconception that foreign-born migrants have deprived South African nationals of employment and other business opportunities, as they felt that migrants are posing a strain on limited social services and amenities, and this has constituted the main drivers of xenophobia [25, 26]. However, the South African government has taken steps to ratify employment laws that are not in favour of migrants. Structural and practical xenophobia scenarios have plunged immigrants into poverty and misery, preventing them from access to all social services. Studies have indicated that undocumented migrants, asylum-seekers and refugees come from diverse African countries that are plagued with endemic and chronic diseases [36, 37], while the burden of non-communicable diseases is found among foreign-born migrants [38–40].

Despite the said reviewed studies, there is a gap in studies and methodological approach that seek to fill the disparities that shows how migrant youths are excluded from using healthcare services. Thus, the primary aim of this study was to document the health services satisfaction and medical exclusion among migrant youths in order to potentially inform future decisions for policy interventions. Therefore, the specific objectives of this study are to: (1) describe the socio-demographics, economic and health-related characteristics by migration status; (2) to determine the prevalence of medical exclusion and health service satisfaction

according to migration status; (3) to examine the factors associated with medical exclusion by migration status; and (4) to examine the predictors of medical exclusion among migrants in Gauteng Province of South Africa. Using a nationally representative datasets allowed the authors to obtain a representative view on migrants' perspectives on the satisfaction of health services; and to investigate the relationships between demographics and medical exclusion practices. Therefore, the rationale for this study is its contribution to an emerging literature that examines health service satisfaction and medical exclusion among migrant youths in South Africa. Hence, its significance, the findings from this study will help to redesign the existing practical interventions that addresses the unmet health needs of youth migrants in South Africa.

## Methods and materials

### Study area

The study area is the South African province called Gauteng, whose name means 'golden place' in Sotho-Tswana. South Africa is the southernmost country in Africa, with a population of over 60 million people and an area of 1,221,037 square kilometres [41]. Gauteng Province is bordered by the Free State, North-West, Limpopo and Mpumalanga Provinces, and it is the smallest province, covering an area of 18 178km$^2$, approximately 1.4% of the total surface area of South Africa, and the most populous being home to 15.8 million people, with a demographically youthful population with a median age of 28 [42, 43]. Geographically, Gauteng lies on the highest part of the interior plateau, on the rolling plains of South Africa's Highveld; its capital is Johannesburg and it also contains the city of Pretoria, as well as the East Rand, West Rand, and Vaal areas [44]. Gauteng Province is divided into three metropolitan municipalities, namely: the City of Tshwane, the City of Ekurhuleni, and the City of Johannesburg, as well as two district municipalities: West Rand and Sedibeng, which are further subdivided into six local municipalities: Mogale City, Rand West City, Merafong, Emfuleni, Midvaal and Lesedi [45, 46]. Gauteng Province is the powerhouse and economic hub of the country and sub-continent, responsible for over 34.8% of the gross domestic product (GDP) as well as being the heart of the commercial, business, and industrial sectors of South Africa. The most important industries, including real estate, business, finance, general government services, and manufacturing services, are located in Gauteng, which is also the financial services centre of Africa. These industries all contribute to Gauteng's GDP [46, 47].

To a great extent, the majority of the foreign banks have their head offices, in Gauteng Province, as well as a number of South African banks, stockbrokers, and insurance giants. However, gold mining constitutes about 80% of Gauteng's mineral production output and the biggest gold and diamond mining houses such as Anglo American and De Beers are situated in Johannesburg, the capital of Gauteng Province [47]. Furthermore, South Africa at present is plagued with persistent droughts and water scarcities which predominantly influence periodic labour migration. Migration within and outside countries in South Africa is driven largely by the pursuit of economic opportunities, political uncertainty and, increasingly, environmental hazards [48, 49]. Thus, industrial developments such as the mining sectors in South Africa, Botswana and Zambia, and the oil wealth of Angola, have been an attractive feature for both skilled and unskilled labour migrants from within the region and elsewhere [14, 45]. According to Statistics South Africa, a net inflow of 852,992 foreign nationals was predicted for the 2016–2020 period, a decrease from the 916,346 forecast for the 2011–2016 period. Five provinces, out of the nine provinces, have experienced a net influx of people moving, including both internal and international migrants. These are Gauteng, the Western Cape, the North-West, Mpumalanga and the Northern Cape [46]. Gauteng has seen the greatest influx of

persons since 2016, and more than three times the numbers seen in the Western Cape province, which had the second higher number of migrants [47, 50]. Also, persons from all provinces, including more than half of the international migrants, are moving to Gauteng, as a result of the economic strength of the province and job opportunities prospects, which has made the province an attractive destination.

## Study design and data source

The Round 5 Gauteng City-Region Observatory (GCRO) Quality of Life (QoL) survey conducted in 2017–2018, a nationally representative survey piloted every two years in South Africa, was utilized in this study. The GCRO engaged a multi-stage and stratified-cluster sampling design, which was used in clustering sampling of households, by random selection of 529 wards utilized as primary sampling units. The Round 5 GCRO's QoL survey was conducted from October 2017 to September 2018 to collect data from the nine municipalities of Gauteng Province. The 2017–2018 GCRO QoL survey data is made available to principal investigators based on written request via the GCRO website (https://www.gcro.ac.za), subsequent to ethical concerns such as voluntary participation, informed consent, respect for persons, anonymity, confidentiality, potential for harm, and communication of research findings. At the sampling level, the first stage was guided by the 2011 South Africa Population Housing Census (PHC) of the definition of enumeration areas (EAs), where EAs were identified and sampled within the selected primary sampling units (PSUs). In the second stage, the cataloguing of households was carried out in each EA sampled, and a sample of households were selected using systematic random sampling.

In addition, all members in each household who met the inclusion criterion (aged 18–29 years) were eligible to participate in the survey. The sample clusters were dispersed across the urban and rural strata within each municipality of the sampled EAs, proportionate to the size of the associated populations within the sampling frame. Clusters (primary sampling units) were assigned to the city and local municipalities within the strata in proportion to the number of households in the census frame for each stratum within the province. Further, survey locations were selected from the dataset of residential dwellings of the 2011 National Census, where respondents were randomly selected by a data collection application called ResearchGo. A sampling frame of all residential structures in Gauteng was achieved using the Geo Terra Image (GTI) and Building-Based Land-Use (BBLU) layer. The Round 5 GCRO QoL 2017–2018 survey collected information from respondents regarding socio-demographic, health-related, socio-economic circumstances, quality of life, attitudes to service delivery, psychosocial attitudes, value-base, and other characteristics. The final samples of the GCRO QoL survey 2017–2018 were 24,889 respondents collected from 52 sampled wards in municipalities of Gauteng province of South Africa. However, the data analyzed in this study were limited to a total of 2,162 immigrants and in-migrants in order to eliminate any potential recall bias. For the 2017–2018 GCRO QoL study, a thorough report on the context, sampling design, questionnaires, sampling frame, data collection techniques, and ethical approval was previously published [51].

## Study population and sample size

Fig 1 below shows the diagram illustrating the stages carried out in the sample size selection of in-migrant and immigrant respondents. A two-year cohort of the GCRO study consisted of 24,889 respondents, out of which 4,872 respondents were sampled according to their assigned categories such as non-migrants, in-migrants, and immigrants. A total of 2,162 respondents involving in-migrants and immigrants were then sampled and utilized as the sample size for

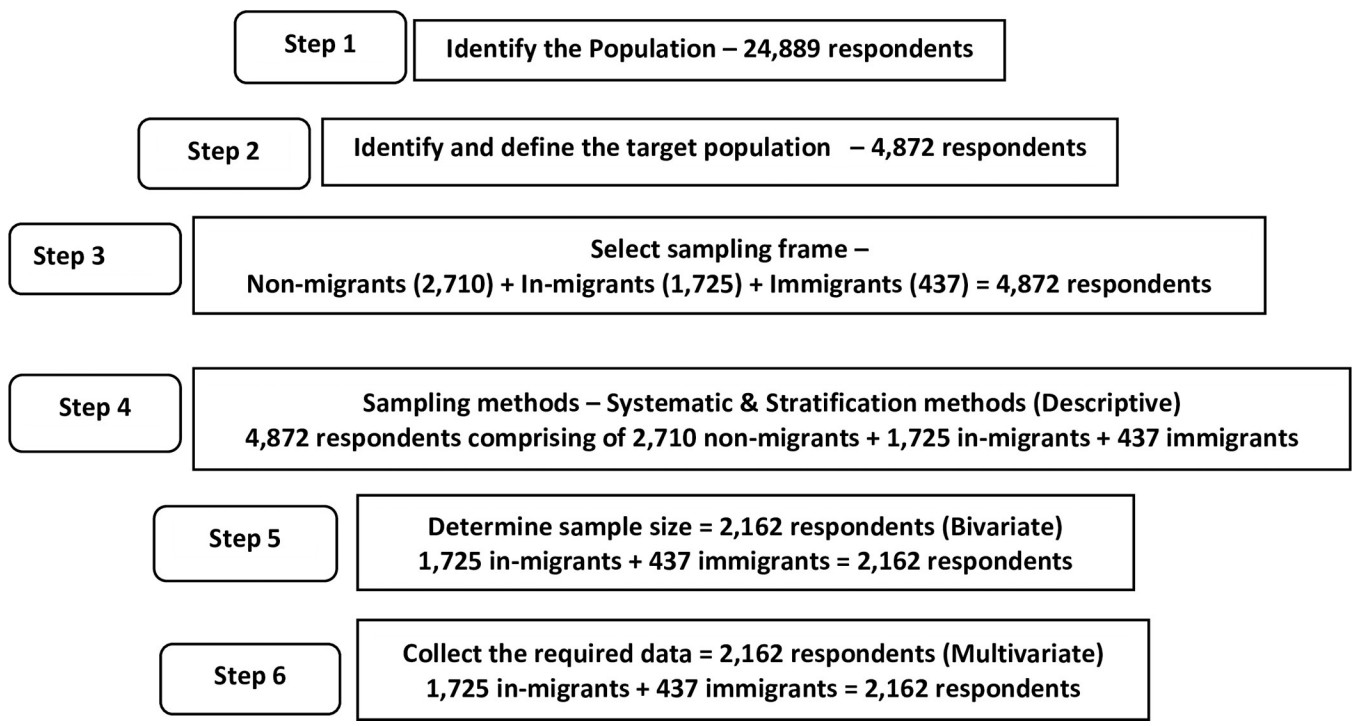

**Fig 1. The Fig 1 [below shows the diagram illustrating the stages carried out in the sample size selection of in-migrant and immigrant respondents.** The diagram illustrated the flow chart of the included and excluded studies and details of sample size replication].

this study. In this study, the population were migrants aged 18–29 years, stratified by 1,725 in-migrants and 437 immigrants, totaling 2,162 (Fig 1). Thus, this study's conceptual clarification and operationalization showed that in-migrants are South African nationals that move within or from one province to another in search of prospective economic opportunities, while immigrants are non-South African nationals who move across international borders for the purposes of economic pursuits and settlement. In order to expand the range of generalization, both migrant groups were used as the target population to increase the level of precision, therefore, justifying its use for the study.

Also, several studies conducted on migration reported a greater influx of in-migrants from rural to urban areas or from non-industrial to industrial areas in search of better job prospects, as this finding has administratively been documented in migration studies conducted in South Africa, hence their large population [3, 34]. Similarly, the population size of immigrants is small as reported by quite a number of studies conducted in South Africa, although this report is not a true reflection of the immigrant population as they are vulnerable to discrimination and exploitation, since the majority of them are poor, illiterate, and reside in slums or shelters [4]. Ineffective policy legislation and administration are not meeting the needs and settlements of immigrants, leading to their poor documentation and insignificant population size for fear of being arrested and deported by the immigration officials in South Africa.

### Variable measurements

**Outcome variable.** The outcome variable for this study was medical exclusion, defined as denial of access or treatment, discretionary health care access, using a derogatory name or words ('*makwerekwere*', a term used to refer to foreigners), negative attitudes and practices of health professionals and employees towards migrants based purely on their identity as non-

South Africans [52, 53]. The outcome variable medical exclusion was re-categorized as 'Yes' (medically excluded, coded as '1') and 'No' (not medically excluded, coded as '0') for use in logistic regression analyses. The 2017–2018 GCRO QoL survey included various questions, including one that asked if anyone in the household needed healthcare in the last year but was unable to obtain it [51]. This was generated from the responses of respondents such as: In the past 12 months, 'was there anybody in this household who needed healthcare but was unable to get it (Q14_05_access)', 'nobody cares about people like me (Q9_11_alienation)', 'what was the main reason that person was unable to get the health care they needed (Q14_06_reason_-health_ behaviour)', and 'what is the main reason that you don't use public health facilities (Q14_02_nonuse_ public_health)'. Thus, the outcome variable is binary in nature and two response categories were created to indicate 'not medically excluded' (No = '0') and 'medically excluded' (Yes = '1'). So 'not medically excluded' category coded as '0' was derived from respondents' responses who said 'no', 'disagree', or 'strongly disagree" to the question asked: 'In the past 12 months, was there anybody in this household who needed healthcare but was unable to get it?' and 'nobody cares about people like me'. The 'medically excluded' category coded as '1' was also derived from responses of respondents who said 'yes', 'agree', 'strongly agree' 'nobody cares about people like me', 'reported being turned away from health facilities', 'did not think it was worth trying to seek care (health care not good enough, health workers' attitudes are bad and thought would get better by oneself)', 'main reason for not using public health facilities', 'what was the main reason that person was unable to get the healthcare they needed?', respondents who indicated 'they have been to public health facilities before and they could not be helped' and 'that the staff are too unfriendly or unhelpful' to the question asked: 'In the past 12 months, was there anybody in this household who needed healthcare but was unable to get it?'

**Independent variables.** We selected independent variables based on the objective of this study and the review of previous studies [54–56], with consideration of the information available in the 2017–2018 GCRO QoL survey. According to the convention used in earlier studies, the variables were broadly divided into three groups: demographics, economic, and health-related factors [6, 40]. Demographic variables assessed in this study included age ('18–19', '20–24' and '25–29'), sex (male* and female), population group (Black African* and Non-Black African), household main spoken language (IsiZulu*, Sesotho, Sepedi and other languages), education (secondary or lower*, matric, and higher) and migration status (in-migrant* and immigrant) [54–56]. Economic variables assessed were employed (no* and yes) and any income (no* and yes). Any income, a proxy for wealth index (socioeconomic status) was derived in the survey through the principal factor of inquiry of available earning accruing over a given period of time (in South African Rands). Health-related factors, including health facility type (private*, public, and both private and public), medical aid cover (yes* and no), health in the last 12 months (excellent*, good, and poor), HIV test in last 12 months (no*, yes, and does not remember), household member HIV status (negative* and positive), disability (no disability* and disabled), mental health condition in the last year (no* and yes) and health satisfaction (satisfied*, dissatisfied, and neither satisfied or dissatisfied) were equally assessed in this study. These variables and their categorization compare well with those of previous studies in South Africa and internationally [57, 58]. (Note: * is the reference category used in analyses).

## Statistical analysis

Prior to data analysis, the dataset was weighted for under-sampling and over-sampling errors based on past studies [59]. In addition, all data analyses were based on migrant status (in-migrants and immigrants) [40]. Univariate analysis was used to describe the characteristics of

the study population against each aforementioned explanatory variable using frequency tabulation, while bivariate analysis, which made use of Chi-Square test, was performed and ρ-values were reported to assess the unadjusted associations between the outcome variable and the various explanatory variables by comparing the differences in the proportion of migrants medically excluded between variable categories. For a bivariate with an independent variable $x$ y and a dependent variable y, the equation is: y = bx + a, *where* y is the dependent variable, $x$ is the independent variable, a is the point where the line of best fit intersects the y-axis and b is the angle of the line. To evaluate the adjusted relationship between outcome and the explanatory variables, multivariable binary logistic regression analyses were carried out, accounting for the effects of all other explanatory variables included in the models. The statistical tests of the binary logistic regression, has the dependent variable, which is a dichotomous (binary) variable, coded as 0 or 1. It specifically helps to determine how much a dependent variable (Y) is affected by one or more independent variables (X), where Y is the dependent variable, X is the independent (explanatory) variable, B is the slope and a is the intercept as well as ε is the residual (error). However, the binary regression model is expressed in terms of the logit instead of $Y : logit = Li = \beta + \beta i X i + \cdots + \beta \kappa X \kappa$. To ensure that no important explanatory factors were missed, variables with ρ≤0.20 in the Chi-Square test were selected for inclusion in the initial multivariable regression model in line with practice in previous studies [60]. This cut-off point was chosen following a critical appraisal of evidence in the literature [38, 40]. A logistic binary regression analysis was then performed to obtain the final close models, which only retained explanatory variables significantly associated with the outcome variable at 5% level (ρ-value < 0.05). Unadjusted and adjusted odds ratios in the close by models together with its 95% CI and ρ-values was reported. To reduce possible statistical errors, analyses were cross-checked, and all variables that satisfied inclusion criterion were included in the models. Multi-collinearity was checked using 'vif' command and the mean vif was 1.57, and data management was performed using Stata. Furthermore, the fixed effects section of the models was made up of demographic-level, economic-level, and health-related level factors. All statistical analyses was conducted using Stata version 14.0 (StataCorp, USA) with the 'svy' command to adjust for sampling weights, clustering effects and stratification. All the regression analyses results were depicted as odds ratios (OR) at 95% confidence intervals (95% CI). All missing values were dropped from the statistical analysis.

## Ethics approval and consent to participate

This study only makes use of secondary data without involving any human subjects. Therefore, no formal ethical approval was required. However, the permission to use the data was sought from the GCRO through a written request. Permission was given subject to using the data for this particular research topic only and publishing the findings in a peer-reviewed journal.

## Results

### Summary of statistics

Household main language, age, education, healthcare facility, health in the past 4 weeks, HIV test in past 12 months, and health services satisfaction, which happens to be key outcome variables that measure medical exclusion were captured as categorical variables (See Table 1).

### Socio-demographic characteristics

Table 2 below shows the demographic, economic and health-related characteristics of non-migrants, in-migrants, and immigrants aged 18–29 years in the Gauteng province of South

**Table 1. Summary statistics of non-migrants and migrants in Gauteng, 2017–2018 (N = 4,872).**

| Social Characteristics | Non migrants | | | | | In-migrants | | | | | Immigrants | | | | |
|---|---|---|---|---|---|---|---|---|---|---|---|---|---|---|---|
| | Weighted observations | Mean | Standard deviation | Min | Max | Weighted observations | Mean | Standard deviation | Min | Max | Weighted observations | Mean | Standard deviation | Min | Max |
| Medical exclusion | 2710 | 0.08 | 0.27 | 0 | 1 | 1725 | 0.06 | 0.23 | 0 | 1 | 437 | 0.04 | 0.20 | 0 | 1 |
| Age | 2710 | 2.37 | 0.69 | 1 | 3 | 1725 | 2.50 | 0.61 | 1 | 3 | 437 | 2.56 | 0.62 | 1 | 3 |
| Sex | 2710 | 1.52 | 0.50 | 1 | 2 | 1725 | 1.51 | 0.50 | 1 | 2 | 437 | 1.45 | 0.50 | 1 | 2 |
| Racial group | 2710 | 1.11 | 0.31 | 1 | 2 | 1725 | 1.03 | 0.17 | 1 | 2 | 437 | 1.06 | 0.23 | 1 | 2 |
| Household main language | 2710 | 2.61 | 1.29 | 1 | 4 | 1725 | 2.86 | 1.22 | 1 | 4 | 437 | 3.26 | 1.21 | 1 | 4 |
| Education | 2710 | 1.95 | 0.70 | 1 | 3 | 1725 | 1.98 | 0.69 | 1 | 3 | 437 | 1.50 | 0.69 | 1 | 3 |
| Health care facility | 2710 | 2.01 | 0.40 | 1 | 3 | 1725 | 2.01 | 0.40 | 1 | 3 | 437 | 1.95 | 0.36 | 1 | 3 |
| Medical aid cover | 2710 | 1.85 | 0.35 | 1 | 2 | 1725 | 1.87 | 0.33 | 1 | 2 | 437 | 1.92 | 0.28 | 1 | 2 |
| Health in the past 4 weeks | 2710 | 1.55 | 0.56 | 1 | 3 | 1725 | 1.60 | 0.55 | 1 | 3 | 437 | 1.61 | 0.52 | 1 | 3 |
| HIV test in past 12 months | 2710 | 1.34 | 0.54 | 1 | 3 | 1725 | 1.34 | 0.57 | 1 | 3 | 437 | 1.43 | 0.59 | 1 | 3 |
| Household member HIV status | 2710 | 1.07 | 0.26 | 1 | 2 | 1725 | 1.05 | 0.22 | 1 | 2 | 437 | 1.03 | 0.17 | 1 | 2 |
| Disability | 2710 | 1.02 | 0.15 | 1 | 2 | 1725 | 1.01 | 0.12 | 1 | 2 | 437 | 1.01 | 0.11 | 1 | 2 |
| HH Mental health condition | 2710 | 1.08 | 0.27 | 1 | 2 | 1725 | 1.06 | 0.24 | 1 | 2 | 437 | 1.03 | 0.16 | 1 | 2 |
| Health services satisfaction | 2710 | 1.71 | 0.89 | 1 | 3 | 1725 | 1.65 | 0.88 | 1 | 3 | 437 | 1.46 | 0.79 | 1 | 3 |
| Employed in the last week | 2710 | 1.25 | 0.44 | 1 | 2 | 1725 | 1.27 | 0.44 | 1 | 2 | 437 | 1.43 | 0.50 | 1 | 2 |
| Any income | 2710 | 1.91 | 0.29 | 1 | 2 | 1725 | 1.93 | 0.26 | 1 | 2 | 437 | 1.93 | 0.26 | 1 | 2 |

Authors' own compilation

Africa. More of the sampled respondents were aged 25–29 years comprising of 63.9% of the in-migrants, and 65.4% of the immigrant population reported medical exclusion, while 50.2% of non-migrants reported non-medically excluded. The population of study had more females reported medical exclusion, in the proportion of 57.8% for the non-migrants and 55.4% of in-migrants, while 51.5% of male immigrants reported medical exclusion. Among the population group, across the non-migrants (93.1%), in-migrants (96.8%) and immigrants (96.9%) who reported medical exclusion were majorly Black African (Table 2).

## Prevalence of medical exclusion according to migration status

Fig 2 shows the prevalence of medical exclusion among in-migrants and immigrants in Gauteng, South Africa. From a total population of 2162 migrants, about 5.8% of in-migrants and 4.2% of immigrants reported having been medically excluded from healthcare services (Fig 2).

## Percentage distribution of the proportion of migrants and the health facility type utilized

Fig 3 shows the percentage distribution of health facility types utilized by migrants in Gauteng province of South Africa. From the sample size of 2,162, more of the sampled in-migrants

**Table 2. Socio-demographic, economic and health-related characteristics of non-migrants and migrants in Gauteng, 2017–2018 (N = 4,872).**

| Demographics characteristics | Non-migrants | | | | In-migrants | | | | Immigrants | | | |
|---|---|---|---|---|---|---|---|---|---|---|---|---|
| | Not Excluded | | Excluded | | Not Excluded | | Excluded | | Not Excluded | | Excluded | |
| | N | % | N | % | N | % | N | % | N | % | N | % |
| | **2492** | **100** | **218** | **100** | **1625** | **100** | **100** | **100** | **418** | **100** | **19** | **97** |
| *Age* | | | | | | | | | | | | |
| 18–19 | 295 | 11.8 | 28 | 13.0 | 101 | 6.2 | 6 | 6.3 | 28 | 6.8 | 2 | 10.2 |
| 20–24 | 947 | 38.0 | 103 | 47.1 | 620 | 38.1 | 30 | 29.7 | 129 | 30.8 | 4 | 21.4 |
| 25–29 | 1251 | 50.2 | 87 | 39.9 | 904 | 55.7 | 64 | 63.9 | 261 | 62.4 | 13 | 65.4 |
| *Sex* | | | | | | | | | | | | |
| Male | 1203 | 48.2 | 92 | 42.2 | 808 | 49.7 | 45 | 44.6 | 229 | 54.9 | 10 | 51.5 |
| Female | 1290 | 51.8 | 126 | 57.8 | 817 | 50.3 | 55 | 55.4 | 189 | 45.1 | 9 | 45.4 |
| *Racial group* | | | | | | | | | | | | |
| Black African | 2207 | 88.6 | 203 | 93.1 | 1578 | 97.1 | 97 | 96.8 | 394 | 94.2 | 19 | 96.9 |
| Non-Black African | 285 | 11.4 | 15 | 6.9 | 47 | 2.9 | 3 | 3.2 | 24 | 5.8 | 0 | 0.0 |
| *Household main language* | | | | | | | | | | | | |
| IsiZulu | 769 | 30.8 | 67 | 30.6 | 420 | 25.8 | 19 | 18.6 | 72 | 17.2 | 5 | 26.3 |
| Sesotho | 447 | 17.9 | 42 | 19.1 | 101 | 6.2 | 4 | 4.4 | 41 | 9.9 | 4 | 17.5 |
| Sepedi | 249 | 10.0 | 25 | 11.5 | 410 | 25.2 | 28 | 28.2 | 2 | 0.4 | 0 | 0.0 |
| Other languages | 1027 | 41.2 | 85 | 38.8 | 695 | 42.7 | 49 | 48.8 | 303 | 72.5 | 10 | 53.1 |
| *Education* | | | | | | | | | | | | |
| Secondary and Lower | 661 | 26.5 | 65 | 29.8 | 392 | 24.2 | 33 | 32.7 | 252 | 60.2 | 14 | 74.6 |
| Matric | 1269 | 50.9 | 118 | 54.2 | 848 | 52.2 | 58 | 57.5 | 120 | 28.7 | 1 | 7.2 |
| Higher | 562 | 22.5 | 35 | 16.0 | 385 | 23.7 | 10 | 9.7 | 46 | 11.1 | 4 | 15.1 |
| ***Economic characteristics*** | | | | | | | | | | | | |
| *Employed in the last week* | | | | | | | | | | | | |
| No | 1856 | 74.4 | 165 | 75.6 | 1185 | 72.9 | 75 | 74.5 | 240 | 57.3 | 9 | 46.9 |
| Yes | 637 | 25.6 | 53 | 24.4 | 441 | 27.1 | 25 | 25.5 | 179 | 42.7 | 10 | 50.0 |
| *Any income* | | | | | | | | | | | | |
| No | 234 | 9.4 | 19 | 8.7 | 119 | 7.3 | 3 | 2.9 | 29 | 6.9 | 3 | 10.1 |
| Yes | 2259 | 90.6 | 199 | 91.3 | 1506 | 92.7 | 97 | 97.1 | 389 | 93.1 | 16 | 86.8 |
| ***Health-related characteristics*** | | | | | | | | | | | | 0.0 |
| *Medical aid cover* | | | | | | | | | | | | 0.0 |
| Yes | 377 | 15.1 | 17 | 7.7 | 212 | 13.1 | 7 | 7.5 | 34 | 8.2 | 2 | 9.9 |
| No | 2116 | 84.9 | 201 | 92.3 | 1413 | 86.9 | 93 | 92.5 | 384 | 91.8 | 17 | 87.0 |
| *Health in the past 4 weeks* | | | | | | | | | | | | |
| Excellent | 1190 | 47.7 | 107 | 49.1 | 715 | 44.0 | 38 | 38.0 | 169 | 40.4 | 9 | 41.9 |
| Good | 1227 | 49.2 | 98 | 44.9 | 863 | 53.1 | 53 | 53.2 | 243 | 58.1 | 10 | 55.0 |
| Poor | 76 | 3.0 | 13 | 6.0 | 47 | 2.9 | 9 | 8.8 | 6 | 1.5 | 0 | 0.0 |
| *HIV test in past 12 months* | | | | | | | | | | | | |
| No | 1728 | 69.3 | 159 | 72.8 | 1149 | 70.7 | 80 | 79.9 | 257 | 61.4 | 12 | 62.4 |
| Yes | 682 | 27.4 | 53 | 24.2 | 395 | 24.3 | 17 | 16.9 | 140 | 33.6 | 7 | 34.5 |
| Does not remember | 82 | 3.3 | 6 | 3.0 | 81 | 5.0 | 3 | 3.2 | 21 | 5.1 | 0 | 0.0 |
| *Household member HIV status* | | | | | | | | | | | | |
| Negative | 2321 | 93.1 | 194 | 89.1 | 1551 | 95.4 | 84 | 83.5 | 406 | 97.2 | 17 | 91.6 |
| Positive | 172 | 6.9 | 24 | 10.9 | 74 | 4.6 | 17 | 16.5 | 12 | 2.8 | 2 | 5.3 |
| *Disability* | | | | | | | | | | | | |
| No disability | 2435 | 97.7 | 208 | 95.7 | 1599 | 98.4 | 100 | 100.0 | 414 | 99.0 | 17 | 89.6 |
| Disabled | 57 | 2.3 | 9 | 4.3 | 26 | 1.6 | 0 | 4.4 | 4 | 0.3 | 2 | 0.0 |

*(Continued)*

**Table 2.** (Continued)

| Demographics characteristics | Non-migrants | | | | In-migrants | | | | Immigrants | | | |
|---|---|---|---|---|---|---|---|---|---|---|---|---|
| | Not Excluded | | Excluded | | Not Excluded | | Excluded | | Not Excluded | | Excluded | |
| | N | % | N | % | N | % | N | % | N | % | N | % |
| | **2492** | **100** | **218** | **100** | **1625** | **100** | **100** | **100** | **418** | **100** | **19** | **97** |
| *HH Mental health condition* | | | | | | | | | | | | |
| No | 2322 | 93.2 | 179 | 82.0 | 1527 | 93.9 | 88 | 88.2 | 407 | 97.3 | 19 | 95.5 |
| Yes | 170 | 6.8 | 39 | 18.0 | 98 | 6.1 | 12 | 11.8 | 11 | 2.7 | 0 | 1.4 |
| *Health services satisfaction* | | | | | | | | | | | | |
| Dissatisfied | 1520 | 61.0 | 65 | 29.7 | 1036 | 63.7 | 38 | 38.3 | 311 | 74.3 | 6 | 30.8 |
| Neither | 306 | 12.3 | 14 | 6.5 | 165 | 10.2 | 8 | 8.2 | 37 | 8.9 | 1 | 5.5 |
| Satisfied | 666 | 26.7 | 139 | 63.9 | 424 | 26.1 | 54 | 53.5 | 70 | 16.8 | 12 | 60.5 |

Source: GCRO QoL Survey, 2017–2018

(84.3%) and immigrants (87.1%) reported the use of a public health facility than other health facility types (private and both health facility types) (Fig 3).

## Percentage distribution of migrants and health service satisfaction

Fig 4 shows the percentage distribution of the proportion of migrants and health service satisfaction in the Gauteng province of South Africa. From a sample of 1,725 in-migrants in the province, about 62.3% reported to be satisfied with the health services they receive while 27.7% indicated to be dissatisfied. Among 437 immigrants, 72.5% indicated to be satisfied with health services they receive, and 18.7% reported being dissatisfied (Fig 4).

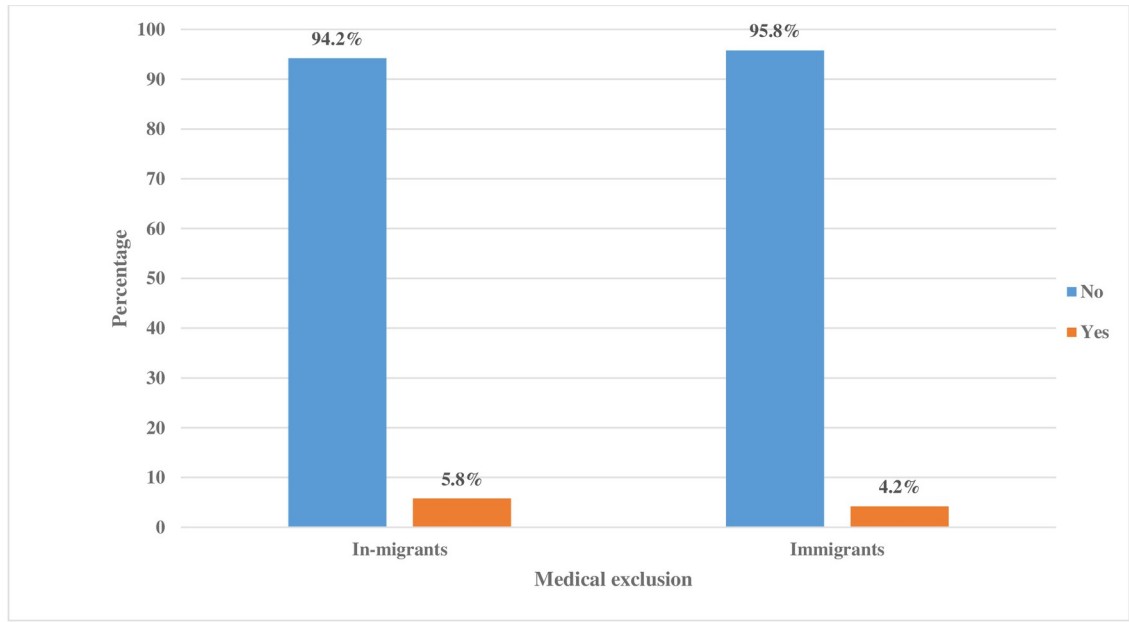

**Fig 2. The Fig 2 [shows the prevalence of medical exclusion among in-migrants and immigrants in Gauteng, South Africa.** The bar chart illustrate the prevalence of medical exclusion among migrants in Gauteng, South Africa. For a representative sample, the prevalence was determined from the number of people in the sample with the characteristic of interest, distributed by the total number of people in the sample size of the migrants].

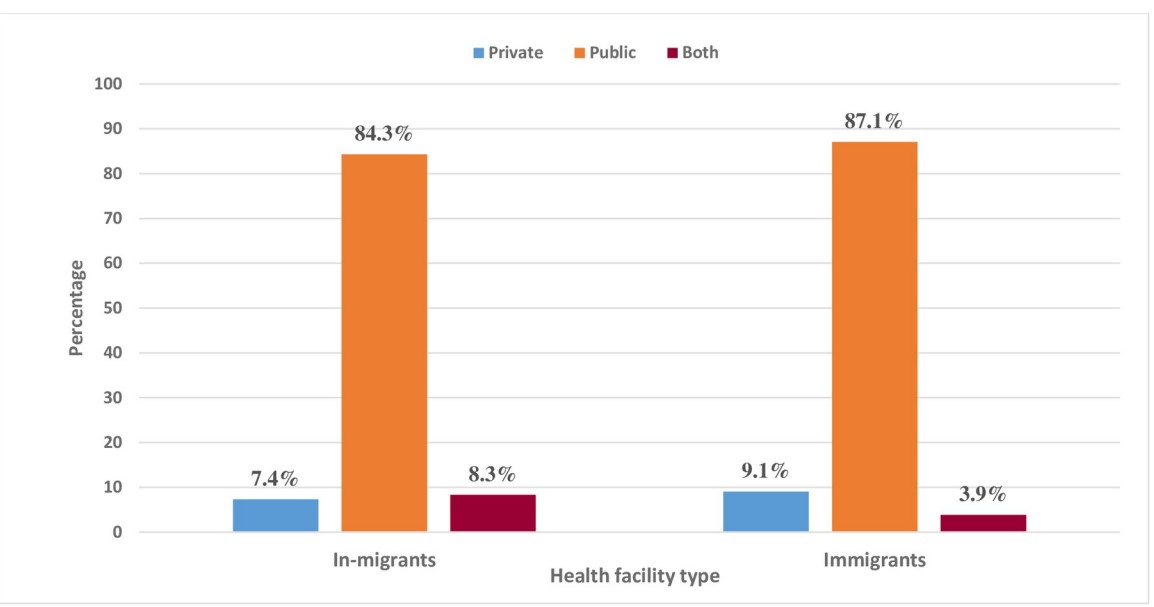

**Fig 3. The Fig 3 [shows the percentage distribution of health facility types utilized by migrants in Gauteng province of South Africa.** The multiple bar chart depict the percentage distribution of utilization of health facility types by migrants in Gauteng, South Africa. The percentage distribution was used to display the 2017–2018 GCRO datasets that indicate the percentage of observations for each data point or the grouping of the data points of the health facility type utilized by migrants].

## Bivariate analysis of factors associated with medical exclusion

From Table 3 below, only age, sex, population group, household main language, education, employed in the last week, any income, medical aid cover, health in the past 4 weeks, household mental health condition and health satisfaction showed to be statistically associated with medical exclusion among in-migrants ($\rho$<0.05). Among immigrants, only age, sex, population group, household main language, education, employed in the last week, having extra income,

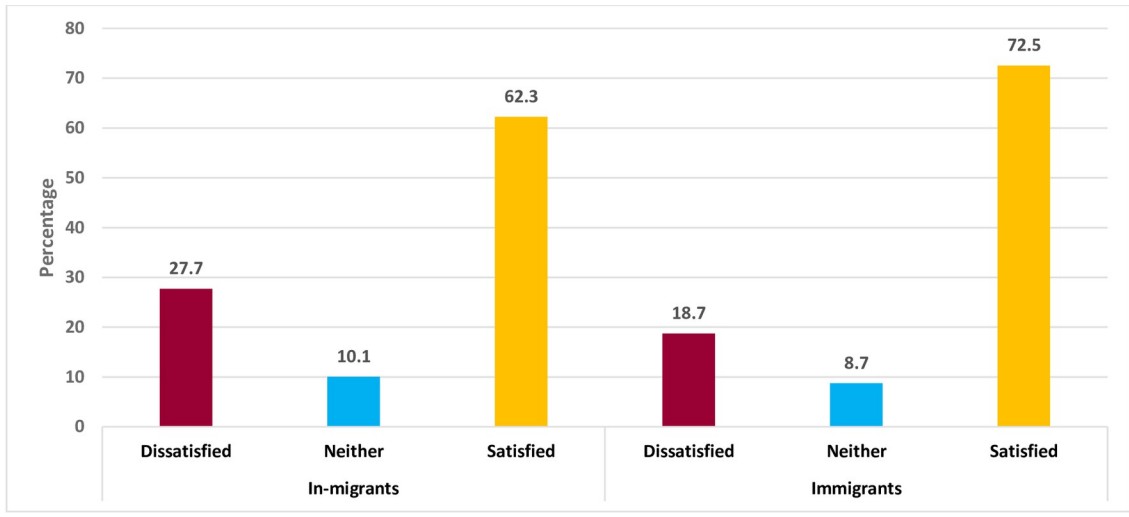

**Fig 4. The Fig 4 [shows the multiple bar chart represent the proportional distribution of health service satisfaction as reported by the migrants in Gauteng, South Africa.** The proportional distribution was used to elicit information on whether migrants are satisfied with the utilization of healthcare services].

medical aid cover, disability and health satisfaction showed to be statistically significant with medical exclusion among in-migrants (ρ<0.05) (Table 3).

## Multivariate analysis of the unadjusted and adjusted logistic regression of the factors associated with medical exclusion

Table 4 showed the results of the multivariate analysis of the unadjusted and adjusted logistic regression analysis. For demographic characteristics, one factor, higher education, is significantly associated with medical exclusion of the migrant population from healthcare services. Similarly, two economic characteristics (employed in the last week and yes to any income), as well as six out of the seven health-related characteristics (no medical cover, poor health in the past four weeks, yes to HIV test in the past 12 months, household member positive HIV status, yes to household mental health condition and dissatisfied with health services) were found to be significantly associated with medical exclusion among migrant populations (ρ< 0.05) (Table 4). Specifically, the odds of medical exclusion of 0.71 are 29% less likely to be reported by the age cohort of 20–24 years compared to those aged18–19 years (Unadjusted Odds Ratio (UOR): 0.71, 95% CI 0.305 – 1.633) while OR of 1.02 were 1.02 times more likely to report medical exclusion among age cohort of 25–29 compared those aged18–19 years (UOR: 1.02, 95% CI: 0.461 – 2.278) (Table 4). In the adjusted odds ratio (AOR), age cohorts of 20–24 years and 25–29 years are both 34% (AOR: 0.66, 95% CI: 0.269 – 1.622) and 6% (AOR: 0.66, 95% CI: 0.269 – 1.622) less likely to report medical exclusion compared to those aged18–19 years, respectively (Table 4).

## Discussion

This study assessed the health services satisfaction and medical exclusion among migrant youths, using the GCRO survey (2017–2018) in Gauteng Province, South Africa. Our study showed a prevalence of 55.6% of non-migrants, 35.4% of in-migrants and 9.0% of immigrants aged 18–29 years in the Gauteng Province in South Africa. This result is higher than the studies conducted at the national level, which showed that only 3% – 4% of people in South Africa are from outside of the country [61]. Also, the prevalence of migrants in this study's findings showed that it is much lower than the prevalence in studies conducted by Statistics South Africa [43] and Mukumbang et al. [4], which reported more than 50% of migrant populations residing in South Africa. In a similar line, the study's findings are lower than those of a survey done in the Southern African Development Community [62], but higher than those of a study done in South Africa, which found a 5% prevalence of migrants with the same characteristics [49]. This study also revealed a prevalence of 6% of in-migrants and 4.2% of immigrant who reported medical exclusion from healthcare services. Thus, this study's findings further showed that about 10% of migrants experienced medical exclusion, with a low prevalence (4.2%) of medical exclusion reported among immigrants. The finding was much lower than other studies conducted in South Africa (20.0% and 48%) [52, 58]. The outcomes were significantly lower than those of research studies conducted in Ghana [63, 64], as well as those conducted in Australia [15] and in Europe [63] combined. The discrepancy might be owing to variations in the study period, that is, Gauteng City-Region Observatory study use GCRO 2017–2018 datasets, while cited studies across Europe and Africa were conducted in 2013, 2018 and 2022.

The disaggregated datasets were stratified by migrant status, and further analysis was carried out to determine the types of health facilities they use. The results indicated a majority of migrant populations (immigrants–87.1% and in-migrants–84.3%) reporting the use of public health facilities rather than other health facilities combined (both private and public). The use

**Table 3. Bivariate analysis of factors associated with medical exclusion among in-migrants and immigrants in Gauteng Province of South Africa, 2017–2018 (N = 2162).**

| Factors | In-migrants | | | | | Immigrants | | | | |
|---|---|---|---|---|---|---|---|---|---|---|
| | Not excluded (n = 1625) | | Excluded (n = 100) | | Chi-square | Not excluded (n = 418) | | Excluded (n = 19) | | Chi-square |
| | N | % | N | % | ρ-value | N | % | N | % | ρ-value |
| **Demographic characteristics** | | | | | | | | | | |
| *Age* | | | | | 0.58** | | | | | 1.77** |
| 18–19 | 101 | 94.1 | 6 | 5.9 | | 28 | 93.6 | 2 | 6.4 | |
| 20–24 | 620 | 95.4 | 30 | 4.6 | | 129 | 96.9 | 4 | 3.1 | |
| 25–29 | 904 | 93.4 | 64 | 6.6 | | 261 | 95.5 | 13 | 4.5 | |
| *Sex* | | | | | 1.89* | | | | | 0.24* |
| Male | 808 | 94.8 | 45 | 5.2 | | 229 | 95.9 | 10 | 4.1 | |
| Female | 817 | 93.7 | 55 | 6.4 | | 189 | 95.6 | 9 | 4.4 | |
| *Population group* | | | | | 1.28* | | | | | 2.66* |
| Black African | 1,578 | 94.2 | 97 | 5.8 | | 394 | 95.5 | 19 | 4.5 | |
| Non-Black African | 47 | 93.7 | 3 | 6.3 | | 24 | 100.0 | 0 | 0.0 | |
| *Household main language* | | | | | 0.99 | | | | | 10.23* |
| IsiZulu | 420 | 95.8 | 19 | 4.3 | | 72 | 93.5 | 5 | 6.5 | |
| Sesotho | 101 | 95.8 | 4 | 4.2 | | 41 | 92.6 | 4 | 7.4 | |
| Sepedi | 410 | 93.6 | 28 | 6.4 | | 2 | 100.0 | 0 | 0.0 | |
| Other languages | 694 | 93.4 | 49 | 6.6 | | 303 | 96.8 | 10 | 3.2 | |
| *Education* | | | | | 7.04*** | | | | | 3.19 |
| Secondary and lower | 392 | 92.3 | 33 | 7.7 | | 252 | 94.7 | 14 | 5.3 | |
| Matric | 848 | 93.6 | 57 | 6.4 | | 120 | 98.9 | 1 | 1.1 | |
| Higher | 385 | 97.5 | 10 | 2.5 | | 46 | 94.2 | 4 | 5.8 | |
| **Economic characteristics** | | | | | | | | | | |
| *Employed in the last week* | | | | | 0.98** | | | | | 0.12** |
| No | 1,184 | 94.1 | 75 | 5.9 | | 240 | 96.4 | 9 | 3.6 | |
| Yes | 441 | 94.5 | 25 | 5.5 | | 178 | 95.0 | 10 | 5.1 | |
| *Having extra income* | | | | | 1.23* | | | | | 1.56** |
| No | 119 | 97.6 | 3 | 2.4 | | 29 | 93.8 | 3 | 6.3 | |
| Yes | 1,506 | 93.9 | 97 | 6.1 | | 389 | 95.9 | 16 | 4.1 | |
| **Health-related characteristics** | | | | | | | | | | |
| *Health facility type* | | | | | 1.91 | | | | | 1.61 |
| Private | 122 | 95.9 | 5 | 4.1 | | 38 | 95.3 | 2 | 4.7 | |
| Public | 1,366 | 94.0 | 88 | 6.1 | | 363 | 95.7 | 17 | 4.4 | |
| Private and public | 137 | 95.3 | 7 | 4.7 | | 17 | 100.0 | 0 | 0.0 | |
| *Medical aid cover* | | | | | 2.85*** | | | | | 0.25*** |
| Yes | 212 | 96.6 | 7 | 3.4 | | 34 | 94.8 | 2 | 5.2 | |
| No | 1,413 | 93.9 | 93 | 6.2 | | 384 | 95.9 | 17 | 4.1 | |
| *Health in the past 4 weeks* | | | | | 15.48** | | | | | 0.54 |
| Excellent | 715 | 95.0 | 38 | 5.0 | | 169 | 95.5 | 9 | 4.5 | |
| Good | 863 | 94.2 | 53 | 5.8 | | 243 | 95.9 | 10 | 4.1 | |
| Poor | 47 | 84.1 | 9 | 15.9 | | 6 | 100.0 | 0 | 0.0 | |
| *HIV test in last 12 months* | | | | | 3.98 | | | | | 1.26 |
| No | 1,149 | 93.5 | 80 | 6.5 | | 257 | 95.6 | 12 | 4.4 | |
| Yes | 395 | 95.9 | 17 | 4.1 | | 140 | 95.5 | 7 | 4.5 | |
| Does not remember | 81 | 96.2 | 3 | 3.8 | | 21 | 100.0 | 0 | 0.0 | |
| *Household member HIV status* | | | | | 27.98* | | | | | 0.37 |

*(Continued)*

**Table 3.** (Continued)

| Factors | In-migrants | | | | | Immigrants | | | | |
|---|---|---|---|---|---|---|---|---|---|---|
| | Not excluded (n = 1625) | | Excluded (n = 100) | | Chi-square | Not excluded (n = 418) | | Excluded (n = 19) | | Chi-square |
| | N | % | N | % | ρ-value | N | % | N | % | ρ-value |
| Negative | 1,551 | 94.9 | 84 | 5.1 | | 406 | 95.9 | 2 | 7.9 | |
| Positive | 74 | 81.8 | 16 | 18.2 | | 12 | 92.1 | 17 | 4.1 | |
| *Disability* | | | | | 1.68 | | | | | 11.19** |
| No disability | 1,599 | 94.1 | 100.0 | 5.9 | | 414 | 96.1 | 17 | 4.0 | |
| Disabled | 26 | 100.0 | 0 | 0.0 | | 4 | 75.9 | 2 | 24.1 | |
| *HH Mental health condition* | | | | | 8.49*** | | | | | 0.65 |
| No | 1,527 | 94.5 | 88 | 5.5 | | 407 | 95.7 | 19 | 4.3 | |
| Yes | 98 | 89.3 | 12 | 10.7 | | 11 | 97.7 | 0 | 2.3 | |
| *Health services satisfaction* | | | | | 48.19** | | | | | 32.66* |
| Neither | 165 | 95.3 | 8 | 4.7 | | 37 | 97.2 | 1 | 2.8 | |
| Dissatisfied | 424 | 88.8 | 54 | 11.2 | | 70 | 85.9 | 12 | 14.1 | |
| Satisfied | 1,036 | 96.4 | 38 | 3.6 | | 311 | 98.2 | 6 | 1.9 | |

**Source:** GCRO, 2017–2018

*ρ < 0.001***

*ρ < 0.01***

*ρ< 0.05* is considered statistically significant (Chi-Square test).*

of healthcare services varies significantly by migrant status, and in South Africa, documented migrant youths are entitled to free medical care to an extent [61, 64]. Studies have shown that immigrants who go to public health facilities will be means-tested in order to check if they qualify for free healthcare and many immigrants have narrated their worst experiences on unfair health treatment they received from healthcare providers [65, 66]. For instance, some immigrants in South Africa reported that their health needs are failing, as these were compounded during the lockdown since they were refused healthcare, and more apprehensions were evident that they might be excluded from the vaccine rollout [67]. Our study also showed that more immigrants than in-migrants reported health services satisfaction as well as dissatisfaction of health services. In contrast, a South African study found that immigrants were less satisfied with the health services they received [68–70]. Similarly, immigrants who reported satisfaction with health services have described uncaring attitudes and perceptions of discrimination when accessing healthcare for the first time. In other studies conducted by Winters et al. [68] and Allegri et al. [69], the uncaring attitudes of health workers towards immigrants pose a huge barrier for exclusion from health care services and a danger to other population groups, especially among immigrants with chronic health conditions such as HIV and TB infections [71, 72].

Among the selected co-variates, demographic characteristics (such as age of respondents as 25–29 years), sex (female) and household main language (Sesotho, Sepedi and other languages)), and health-related characteristics (such as no medical aid cover, health in the past four weeks (good and poor), household member HIV status (positive), household mental health condition (yes), and health services satisfaction (neither satisfied nor dissatisfied) were the major predictors of medical exclusion among migrants in the unadjusted and adjusted logistics regression analysis. We also found a non-significant relationship between age of respondents and medical exclusion among migrants in South Africa. The unadjusted odds ratio showed that respondents who are aged 25–29 years were more likely to be medically

**Table 4. Unadjusted and adjusted logistics regression analysis of predictors of medical exclusion among migrants in Gauteng Province of South Africa, 2017–2018 (n = 2162).**

| Factors | Unadjusted | | | Adjusted | | |
|---|---|---|---|---|---|---|
| | Odds ratio | 95% CI | ρ-value | Odds ratio | 95% CI | ρ-value |
| *Demographics characteristics* | | | | | | |
| *Age* | | | | | | |
| 18–19 (RC) | | | | | | |
| 20–24 | 0.71 | 0.305 – 1.633 | 0.42 | 0.66 | 0.269 – 1.622 | 0.37 |
| 25–29 | 1.02 | 0.461 – 2.278 | 0.95 | 0.94 | 0.398 – 2.221 | 0.89 |
| *Sex* | | | | | | |
| Male (RC) | | | | | | |
| Female | 1.21 | 0.781 – 1.888 | 0.39 | 1.07 | 0.678 – 1.705 | 0.76 |
| *Population group* | | | | | | |
| Black African (RC) | | | | | | |
| Non-Black African | 0.76 | 0.181 – 3.235 | 0.72 | 0.89 | 0.202 – 3.892 | 0.87 |
| *Household main language* | | | | | | |
| IsiZulu (RC) | | | | | | |
| Sesotho | 1.13 | 0.510 – 2.490 | 0.77 | 1.06 | 0.464 – 2.409 | 0.89 |
| Sepedi | 1.43 | 0.742 –2.737 | 0.29 | 1.42 | 0.709 – 2.835 | 0.32 |
| Other languages | 1.23 | 0.745 – 2.032 | 0.42 | 1.42 | 0.840 – 2.395 | 0.19 |
| *Education* | | | | | | |
| Secondary and Lower (RC) | | | | | | |
| Matric | 0.84 | 0.524 – 1.334 | 0.45 | 0.77 | 0.456 – 1.287 | 0.31 |
| Higher | 0.40 | 0.197 – 0.819 | 0.01** | 0.36 | 0.163 – 0.815 | 0.01** |
| **Economic characteristics** | | | | | | |
| *Employed in the last week* | | | | | | |
| No (RC) | | | | | | |
| Yes | 0.96 | 0.577 – 1.608 | 0.00*** | 0.94 | 0.556 – 1.603 | 0.03* |
| *Any income* | | | | | | |
| No (RC) | | | | | | |
| Yes | 0.84 | 0.837 – 4.022 | 0.01** | 0.36 | 0.971 – 5.730 | 0.05* |
| **Health-related characteristics** | | | | | | |
| *Medical aid cover* | | | | | | |
| Yes (RC) | | | | | | |
| No | 1.60 | 0.702 – 3.652 | 0.00*** | 1.23 | 0.450 – 3.362 | 0.05* |
| *Health in the past 4 weeks* | | | | | | |
| Excellent (RC) | | | | | | |
| Good | 1.11 | 0.714 – 1.723 | 0.65 | 1.09 | 0.691 – 1.724 | 0.71 |
| Poor | 3.21 | 1.357 – 7.588 | 0.01** | 1.89 | 0.745 – 4.805 | 0.01** |
| *HIV test in past 12 months* | | | | | | |
| No (RC) | | | | | | |
| Yes | 0.67 | 0.375 – 1.193 | 0.05* | 0.74 | 0.397 – 1.381 | 0.02* |
| Does not remember | 0.48 | 0.116 – 2.021 | 0.32 | 0.49 | 0.122 – 1.960 | 0.31 |
| *Household member HIV status* | | | | | | |
| Negative (RC) | | | | | | |
| Positive | 3.96 | 2.181 – 7.203 | 0.00*** | 3.67 | 1.971 – 6.819 | 0.00*** |
| *Disability* | | | | | | |
| No disability (RC) | | | | | | |
| Disabled | 0.79 | 0.161 – 3.868 | 0.77 | 0.42 | 0.087 – 2.045 | 0.28 |

*(Continued)*

**Table 4.** (Continued)

| Factors | Unadjusted | | | Adjusted | | |
|---|---|---|---|---|---|---|
| | **Odds ratio** | **95% CI** | **ρ-value** | **Odds ratio** | **95% CI** | **ρ-value** |
| *HH Mental health condition* | | | | | | |
| No (RC) | | | | | | |
| Yes | 1.99 | 1.024 – 3.869 | 0.04* | 1.55 | 0.789 – 3.031 | 0.02* |
| *Health services satisfaction* | | | | | | |
| Satisfied (RC) | | | | | | |
| Neither | 1.39 | 0.563 – 3.425 | 0.48 | 1.59 | 0.606 – 4.174 | 0.35 |
| Dissatisfied | 4.01 | 2.534 – 6.345 | 0.00*** | 4.29 | 2.528 – 7.270 | 0.00*** |

*Source*: GCRO Survey, 2017–2018; Significant p-values

*ρ ≤ 0.05

** ρ ≤ 0.01

*** *ρ ≤ 0.001*: 95% Confidence intervals (CI); AOR, adjusted odds ratio; UOR, unadjusted odds ratio; RC, Reference Category; Adjustment variables of the multivariable models are age, marital status, educational level, residence, work status, wealth quintile, and provinces.

excluded from health care services compared to those who are aged 18–19 years. This could be due to the stigmatization that goes with age group of 25–29 years, that they are the major migrant populations that are over-burdening the health care facilities in South Africa. Studies of a similar nature conducted in sub-Saharan Africa demonstrated that as migrants aged, they were more likely to be denied access to medical care services [73–75]. Our study revealed that, compared to their male counterparts, female migrants were more likely to be denied access to healthcare services. This clearly illustrates the gender gap in accessing healthcare facilities. Women encounter many healthcare hurdles, which prevents them from easily accessing and acquiring the necessary care. A recent report revealing how a Limpopo Provincial Health Minister reprimanded a Zimbabwean female migrant who was seeking medical treatment in South Africa, leading to a protest against medical xenophobia [64, 65]. Another well-cited instance indicated how a South African advocacy group for the immigrant population launched court proceedings against an upfront fee of an estimated 935 US Dollars for maternity cases, and 3000 US Dollars for routine surgical cases for female migrants, while non-migrants with such cases have free access to all aforementioned treatment [64, 65].

We also found a significant relationship between the economic characteristics and medical exclusion among migrants in both unadjusted and adjusted odds ratio. Our study showed that migrants who were employed in the last week were less likely to be medically excluded from healthcare services in both models, while migrants with any income were found less likely to be medically excluded from healthcare services in both models. This aligns with a study conducted in South Africa that migrants pay for their healthcare services just as South African nationals do [64]. Thus, non-South Africans are either subject to the same means-tested hospital fees, or they are subject to the highest fees if they are undocumented and not from the Southern African Development Community (SADC) [52, 62] and this was also shown in a recent study conducted in Europe [63, 76]. Consequently, the out-of-pocket payment system for healthcare services is a significant obstacle for unemployed migrants who lack any means of supplemental income. This typically will have an impact on the most vulnerable and destitute migrants who cannot afford to pay for their medical expenses [77, 78].

Regarding the health-related characteristics, we found that the odds of migrants who were medically excluded were higher among those with no medical aid cover in both models. Studies based on migrants who did not have medical aid, thereby experiencing medical exclusion, had reported similar findings [77, 78]. Similarly, migrants with poor or good health in the past

4 weeks had higher odds of being medically excluded from healthcare services in both unadjusted and adjusted models. A possible explanation could be the two health-related characteristics at migrants awaiting clarification of their status and those without documentation will experience some forms of medical xenophobia [65, 78]. The findings show that the odds of migrants who were medically excluded decreases with those who had HIV test in the past 12 months. Given that South Africa has an estimated TB incidence of 860/100,000 and an estimated HIV + TB co-infection of 520/100,000, this could be one explanation for the lack of attention given to migrants living with HIV/TB or/and co-infections [79]. Despite this, TB is still the biggest cause of death for people living with HIV in South Africa, and the rise of drug-resistant TB strains, which are more challenging and expensive to treat, has made the situation even worse. In order to expedite the start of antiretroviral therapy (ART) for PLHIV with TB, the Centers for Disease Control and Prevention (CDC) in South Africa, through its Global AIDS Program (GAP), is collaborating closely with the NDoH, much like with Covid-19, to strengthen HIV/TB screening for all persons living with HIV (PLHIV), including migrants [80].

The study also suggests that a household member with a positive HIV status predicts medical exclusion of migrants from healthcare services. In the unadjusted and adjusted models, migrants who had a household member with a positive HIV status were 4.0 and 3.7 times more likely to experience medical xenophobia, respectively. Similar to the aforementioned finding, studies conducted in Mozambique [81] and South Africa [82] reported conclusions that were similar. One of the possible explanations could be that there is a gap in literature on barriers that prevent migrants from reporting their household member with HIV-positive status or engaging healthcare systems with such medical problems. Also, shifting cultural and clinical settings may result in structural vulnerabilities that are limiting immigrants' household members with such medical history of HIV-positive status from accessing and having proper integration within healthcare services [80, 81]. However, these barriers include stigmatization of HIV-positive household members, social seclusion, xenophobia and deportation, marginalization and mistreatment, language obstacles, ethnic hostility, and medical heterogeneity [81, 82]. Simultaneously, these barriers may lead to medical exclusion, deferment of treatment-seeking and deterring drug adherence, which could escalate proportions of indisposition and death as well as promoting viral mutation and antiretroviral drug resistance [83].

Our study findings also suggest that more migrants with a household member with a mental health condition were reported to have experienced medical exclusion from health services compared to those who do not have a household member with a mental health conditions. Some studies have reported that immigrants with a household member with mental health challenges could face experienced, anticipated, and internalized stigmas, from stereotyping and prejudice to discriminatory attitudes from healthcare providers [77, 78], and so will not be able to communicate their household health problems to the public. Hence, we found that the predictors for migrants who were neither satisfied nor dissatisfied with health services were reported to be medically excluded from health care services. Similar studies conducted in Lesotho [80] and in Zimbabwe [83] have reported results similar to this study's findings. The possible explanation could be that migrants who were excluded from the health facilities initially may report high rates of being neither satisfied nor dissatisfied with the health services and health providers [81].

## Implications of findings on medical exclusion of migrants in South Africa

The violation of migrants' rights to access health care has grave consequences, having a gendered, population group and class influence, with poor and economically disadvantaged

immigrants bearing the burden of this discrimination [61, 65]. Undocumented immigrants, refugees, and those seeking asylum may have been exposed to communicable diseases during their long journey to South Africa from their home countries. In order to tackle migration and health beyond infectious diseases and border checks, the South African government and pertinent health stakeholders should establish a comprehensive multi-sectoral approach. The Immigration Act must be revised to properly recognize the rights to healthcare of both documented and undocumented immigrants [64, 67]. Providing better care includes some understanding of migrants' medical concerns within their social context, particularly culture of origin and the challenges of migration can be included in healthcare policies or interventions targeting migrant youths. The law should be supplemented by an extensive national strategy that specifies how undocumented migrants should be handled and it should be administered uniformly throughout all provinces. Also, to enforce migrant health rights, we must speak up and educate health professionals. Such training, developed in collaboration with the South African Department of Health and the Health Professions Council of South Africa, ought to raise and sensitize the awareness of rights and requirements for health care for migrants among medical practitioners. Health administrators should be a part of it as well, as they serve as a point of entry for immigrants trying to receive medical services [65, 84]. The study findings displayed that these measures are necessary, as a public healthcare system that excludes migrants creates conditions for poor public health for all. It increases the vulnerability of migrants, generates and magnifies discrimination and inequalities in health, and violates migrants' constitutional rights to access health care. Furthermore, this study findings showed that it is not just a health and human rights issue but it is also a matter of social justice. In South Africa's society and economy, migrant labour has played a crucial role, and their lower wages have increased consumer and business profitability and saved consumers money. Hence, delivering equitable access to care for migrants can reduce the health and social costs of disease, improve social cohesion, protect public health and human rights, and contribute to healthier migrants in healthier local communities.

## Strengths and limitations

This study presents a snapshot of medical exclusion and satisfaction rates of health services among migrant youths in Gauteng province of South Africa. To the best of our knowledge, this is the first study to investigate health services' satisfaction and medical exclusion among youth migrants using nationally representative data in South Africa. National representativeness, high response, application of complex sample statistics in all analyses (to adjust for sample weights and cluster design of the survey) and low missing data are some of the strengths of this study. Others include large sample size and the use of a migrant status data disaggregation method. A methodological strength is the combination of univariate, bivariate and multivariate analyses to explore whether medical exclusion from healthcare services can facilitate satisfaction of migrants by status exacerbates exclusion for migrants in South Africa. Regarding the study's limitations, due to the cross-sectional nature of its design, it is possible that the findings will not prove a true causal relationship between the independent variables and the outcome variable. The respondents' self-reports from the two years prior to the 2017–2018 GCRO survey were used to compile the data, which raises the possibility of recall bias and misclassification bias. The low prevalence of immigrant populations, especially for youth cohorts, remains less visible, yet a reflection of a typical pattern of temporary migration (migrant worker) remains insufficiently understood. More research is needed to confirm and explore this topic, hence future studies may consider addressing this limitation.

## Conclusion

This paper, based on the data from the fifth round of the GCRO, contributes new evidence to improve our understanding of the health services satisfaction and association with medical exclusion through the analysis of socio-demographic, economic and health-related determinants amongst in-migrants and immigrants in South Africa. In-migrants were found to have reported higher prevalence of medical exclusion from health services (5.8%) with lower health services satisfaction (37.8%) than immigrants. These findings, which will be enhanced in future longitudinal follow-up rounds, offer important insights into how migrant youths interface with health service satisfaction in a transitioning context such as South Africa. As such, the study assists in providing evidence to support the redesigning of health policies that will cover the effective healthcare for all migrants as part of achieving the sustainable development goal 3 and the universal health coverage. Also, the unmet needs of the South Africa's sizeable immigrant community should be addressed through policies reforms that will identify these vulnerable groups through political and health-systems intervention provided. Thus, legislative changes that is free of corruption can be employed to track the public health financing and strategize to improve the performance and management of the health system that will ensure a prevention and mitigation of medical exclusion of in-migrants and immigrants from healthcare services in South Africa.

## Recommendation

Based on the data and findings, the authors provide the following recommendations: first, our findings call for the need to implement health programmes among migrants in South Africa to increase awareness of the negative implications of medically excluding migrants from health services, in order to avoid health risks of morbidity complications and mortality. Second, health legislation and policies should be formed to focus and shape health insurance coverage and, ultimately, access to and utilization of healthcare services among migrants in South Africa. Third, there should be an urgent need for revision of an enabling constitution, national health care act, and an exclusionary immigration act, as well as an NHI bill in order to ensure an inclusive healthcare system for all people, regardless of their nationality, language spoken, and social status. Fourth, South Africa should work to develop a national migration and health-coordinating network and policy, by drawing on existing policy processes at the local and national level, and in consultation with multiple stakeholders. Findings from this study may be useful in informing policy-makers and public health experts in this area so as to improve the health outcomes of migrants, by improving the utilization of health facilities, especially among female migrants and those vulnerable migrants with chronic health conditions such as HIV and mental health issues.

## Supporting information

**S1 File. The online version contains supplementary material and the Do-file for the analysis is uploaded at the time of submission.**
(PDF)

## Acknowledgments

The authors are grateful to GCRO surveys for providing them with the access to the data set. We are also thankful to Mrs. Helen Thomas for her support in language editing.

## Author Contributions

**Conceptualization:** Monica Ewomazino Akokuwebe, Salmon Likoko.

**Data curation:** Monica Ewomazino Akokuwebe, Salmon Likoko.

**Formal analysis:** Monica Ewomazino Akokuwebe, Salmon Likoko.

**Funding acquisition:** Godswill Nwabuisi Osuafor.

**Investigation:** Monica Ewomazino Akokuwebe, Salmon Likoko.

**Methodology:** Monica Ewomazino Akokuwebe.

**Project administration:** Monica Ewomazino Akokuwebe.

**Software:** Salmon Likoko.

**Supervision:** Monica Ewomazino Akokuwebe, Godswill Nwabuisi Osuafor, Erhabor Sunday Idemudia.

**Validation:** Monica Ewomazino Akokuwebe.

**Writing – original draft:** Monica Ewomazino Akokuwebe.

**Writing – review & editing:** Monica Ewomazino Akokuwebe, Godswill Nwabuisi Osuafor, Salmon Likoko.

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
