## [Decision Letter · Decision Letter 0]

21 Feb 2023

PONE-D-22-33773Health Services Satisfaction and Medical Exclusion among Migrant Youths in Gauteng Province of South Africa: A Cross-sectional Study of the GCRO Survey (2017−2018)PLOS ONE

Dear Dr. Akokuwebe,

Thank you for submitting your manuscript to PLOS ONE. After careful consideration, we feel that it has merit but does not fully meet PLOS ONE’s publication criteria as it currently stands. Therefore, we invite you to submit a revised version of the manuscript that addresses the points raised during the review process.

We look forward to receiving your revised manuscript.

Kind regards,

Engelbert Adamwaba Nonterah, MD, PhD

Academic Editor

PLOS ONE

Journal Requirements:

2. Please note that in order to use the direct billing option the corresponding author must be affiliated with the chosen institute. Please either amend your manuscript to change the affiliation or corresponding author, or email us at plosone@plos.org with a request to remove this option.

3. Please upload a new copy of Figures 1 to 5 as the detail is not clear. Please follow the link for more information: https://blogs.plos.org/plos/2019/06/looking-good-tips-for-creating-your-plos-figures-graphics/

4. We note that Figure 1 in your submission contain map images which may be copyrighted. All PLOS content is published under the Creative Commons Attribution License (CC BY 4.0), which means that the manuscript, images, and Supporting Information files will be freely available online, and any third party is permitted to access, download, copy, distribute, and use these materials in any way, even commercially, with proper attribution. For these reasons, we cannot publish previously copyrighted maps or satellite images created using proprietary data, such as Google software (Google Maps, Street View, and Earth). For more information, see our copyright guidelines: http://journals.plos.org/plosone/s/licenses-and-copyright.

Reviewers' comments:

Reviewer's Responses to Questions

**Comments to the Author**

1. Is the manuscript technically sound, and do the data support the conclusions?

Reviewer #1: Yes

Reviewer #2: Partly

2. Has the statistical analysis been performed appropriately and rigorously? 

Reviewer #1: Yes

Reviewer #2: No

3. Have the authors made all data underlying the findings in their manuscript fully available?

Reviewer #1: Yes

Reviewer #2: Yes

4. Is the manuscript presented in an intelligible fashion and written in standard English?

Reviewer #1: Yes

Reviewer #2: No

5. Review Comments to the Author

Reviewer #1: The manuscript is well written but it needs minor revisions.

A detailed review report is uploaded as attachment.

The authors did not clarify whether this study was cleared by any ethical review committee.

Reviewer #2: February 17,2023

Dear Editor,

Thank you for the opportunity to review this manuscript titled: “Health Services Satisfaction and Medical Exclusion among Migrant Youths in Gauteng Province of South Africa: A Cross-sectional Study of the GCRO Survey (2017−2018)”. The study presents an important health inequity issue, using a relatively large sample. However, I have issues with the write-up and results presentation. One major observation is that the background majorly focused on immigrants (foreign nationals) but the sample upon which the analysis was conducted were immigrants (foreign nationals who migrated to the Country) and in-migrants (South African nationals who moved within the Country, a large part of the sample). These different classes of migrants have their peculiar situations and so different concerns or rules will apply to them. Unfavorable policies or health workers’ position towards South Africans, irrespective of whether they migrate to a new Province would differ from that of foreign nationals. Therefore, I have issues with the authors not clearly delineating these differences in their background, and discussion.

See below for detailed comments in relation to different sections of the article:

Abstract

The authors utilized several concepts, many of which were not predefined or explained in the abstract which made it hard to read and follow. The abstract is the first encounter readers will have with an article and if it is very unclear, they may lose interest.

- For instance, in the abstract background, they mentioned medical exclusion (the major outcome of the study), and used other terms like medical mistreatment, social phobia and health services satisfaction. I will suggest that the authors avoid shopping around words and should stick to and focus the background on the primary outcome of interest.

- Also, the study population were migrants which the authors later classified into in-migrants and immigrants, and presented the results using these terms. The authors should consider defining them in the abstract for clarity.

Background

Fair.

As earlier mentioned, the background to the study majorly focused on immigrants (foreign nationals), why they immigrate and the impact on their health and the need to access healthcare. While this is clear, the actual study sample involved in-migrants (SA nationals). SA national migrants (in-migrants) who face medical exclusion may be due to completely different issues which were not touched on in the background at all. A lot of the information in the background do not relate to the in-migrants. I suggest you include this important, yet missing information.

Paragraph 2 is quite lengthy, and many sentences were basically repeating the same thing or unnecessary information. Consider shortening this and keeping information that are relevant to the outcome of interest.

In paragraphs 3 and 4, the authors already highlighted several factors that impact healthcare utilization and access, backed with several citations. Given that, the authors did not provide a proper justification for the current study. The authors stated that there are gaps on how youths are prevented from healthcare services but I don’t think this study address that. Likewise, they stated that they wanted to predictors of healthcare satisfaction and medical exclusion, whereas, the only outcome measured in the study was medical exclusion and satisfaction was merely a predictor. Consider revising your objective or addressing the gap in your result.

Methods

Well written and detailed enough.

Results

- In the Tables, use n instead of F or write Frequency.

- Place the text describing Table 1 before the Table.

- Consider adding where the result in Figure 2 can be presented. That is, the frequency and percentage of non-immigrants, immigrants and in-migrants. You do not need this information on a separate Figure.

- The results descriptions are too detailed. The results are already presented in the Tables, describe the interesting findings, rather than describing all the results in the Table, especially the results for Tables 1 and 3 (extremely long).

- Rectify the results describing Table 2. Comment in text.

- Some of the frequencies when added up, exceeds the total frequencies for that group. E.g. employment status for the not excluded group was equal to 1626 but you reported the overall sample is 1625. Please make sure all the results presented are correct.

- The description of results on Table 3 is extremely long and the odds ratios were mostly interpreted incorrectly or unclearly. If you have an OR of >1, that is increased odds and OR<1 is decreased odd. Categories with say OR of 0.70 are 30% less likely while OR of say 3.22 will be 3.22 times more likely. Please present interesting findings with clear and accurate interpretations.

-The use of UAOR (Unadjusted AOR) is incorrect. You either have an adjusted OR or unadjusted OR. AOR means adjusted odd ratio.

- All the figure are very blurry. Please remake. Consider deleting Figures 1 and 2. Too many information in the manuscript already and Figures seems redundant.

Discussion, recommendation, implications

- The authors had a very lengthy discussion. The discussion is the “so what” of the study. It should focus on the key take away points, the meaning, the potential reasons underlying them, how they compare with other studies and the implications.

- The authors did not properly discuss the results for the different groups and the implications. What you would expect among SA national migrants would be different from foreign nationals and proper discussion regarding that should be made. For now, the authors have merged the two classes of migrants (immigrants and in-migrants), who are clearly different and will face different challenges.

- Aside the lengthy discussion, the authors also had lengthy implications and recommendations. This is a research article and not a thesis. Try to summarize and drive home the key points.

Conclusion

- The conclusion should be based on the findings.

- The authors added the prevalence for both groups and indicated the prevalence was 10% and very high. This is in relation to what?

References

- This is not a review paper. The references are just so many. Delete redundant references.

Minor comments

Title: I suggest you change “A cross-sectional study of the GCRO survey” to “A cross-sectional analysis of the GCRO survey” rather than “A cross-sectional study of GCRO survey”. Also, consider spelling out GCRO.

Keywords: Factor is not needed as a keyword

Abstract

- Dissatisfaction of healthcare services should be changed to dissatisfaction with health care services”

Background

- Consider revising this from “Regardless of their gender, they are often victims of violence, infectious diseases, and malnutrition….” to “Regardless of gender, migrants are often victims of violence, infectious diseases, and malnutrition…”. The excessive use of “they” in the paragraph makes it challenging to follow.

- Line 8: The word “ordinary” should be removed “…ordinary non-migrant” as the meaning of ordinary in that context wasn’t clear.

- The latter part of this sentence is unclear “ The trend and pattern of migration ranges from low-income to high-income nations, including those African countries with political, social and economic stability, envisages a growing prevalence of migrant youths”. Consider revising.

- Before ref 15, change migrant people to migrants.

Other minor comments were indicated in the manuscript.

6. PLOS authors have the option to publish the peer review history of their article (what does this mean?). If published, this will include your full peer review and any attached files.

Reviewer #1: No

Reviewer #2: No

---

## [Author Response · Author response to Decision Letter 0]

17 May 2023

1st Reviewer’s Responses 1 #

Health Services Satisfaction and Medical Exclusion among Migrant Youths in Gauteng Province of South Africa: A Cross-sectional Study of the GCRO Survey (2017−2018)

Title

• The title is clear and precise.

Abstract

-In the abstract and the introduction, the authors summarized the main research question and key findings.

-There is no mention of health services satisfaction in the aim. [health services satisfaction has been included - Page 1]

This study aimed to determine the prevalence and factors contributing to medical exclusion among migrant youths in Gauteng Province, South Africa. [health services satisfaction has been included - Page 1]

-Please consider shortening your sentences. The second sentence under the method subtitle is very long. [To an extent, it has been reduced to be able to retain its same meaning - Page 1]

-To reduce the word count the authors can just mention ρ≤0.05 (it covers the other two specified p-values).

 “At the bivariate level, demographic (age, sex, and population group), economic (employed and any income) and health-related (no medical aid and household member with mental health) factors were significantly associated with medical exclusion (ρ≤0.05, ρ≤0.01 ρ≤0.001)”. [Corrected in red ink - Page 1]

-Please consider paraphrasing the sentence below, it doesn’t read well.

“The adjusted odds ratio (AOR) of logistic regression indicated factors such as female gender (AOR 1.07, 95% CI 0.678, 1.705), no medical aid cover (AOR 1.23, 95% CI 0.450‒3.362), and neither (AOR 1.59, 95% CI 0.606‒4.174) or dissatisfied (AOR 4.29, 95% CI 2.528‒7.270) with the health services were predictors of medical exclusion”. [Corrected in red ink - Page 1]

Introduction

-Please provide some migration statistics on what is happening currently in South Africa to give a clear picture of what is currently happening. [Addressed in red ink - Page 3]

-Please elaborate, give examples of these factors, where were these studies conducted?

Factors associated with migrants’ exclusion from healthcare access have been cited in several studies [20-22]. [Addressed in red ink - Page 3]

-Please cite these studies

Also, other studies have mentioned medical xenophobia as one of the major barriers that migrants are faced with and this has been a hindrance to healthcare accessibility in South Africa. [Addressed in red ink - Page 3]

Methods

-This sentence can be moved to the introduction. 

In addition, SA Statistics (2020) reported an estimate of 2.9 million migrants who are presently residing in South Africa at mid-year 2020 [7]. [Has been moved to the introduction and addressed in red ink - Page 4]

-Avoid repetition [Addressed in the entire manuscript]

-Describe the study design of the current study (not the GCRO) [This study is a secondary study where the data used was collected from the GCRO database. However, secondary data do not have their own study design, as they are not the primary collectors of the GCRO data. Note that secondary research is a research method that involves using already existing data. Existing data is summarized and collated to increase the overall effectiveness of the research. Therefore, the researchers use the already existing data also known as secondary data, and this existing data is then summarized and arranged to increase the overall efficacy of the study. In this study, the study design was adopted from the GCRO study design and was utilize to suit the methods and analytical framework of this study] {Look at these studies on the aforementioned - 1. Baldwin JR, Pingault JB, Schoeler T, et al. Protecting against researcher bias in secondary data analysis: challenges and potential solutions. Eur J Epidemiol., 2022; 37, 1-10; 2. Tripathy JP. Secondary data analysis: Ethical issues and challenges. Iran J Public Health. 2013 Dec; 42(12): 1478-9}.

-It is not very clear how the authors got the sample size or how the potential recall bias was eliminated. A flow diagram on how the sample size was obtained would have been helpful.

However, the data analyzed in this study were limited to a total of 2,162 immigrants and in-migrants in order to eliminate any potential recall bias {A flow diagram on sample size has been inserted - Page 4}

Results

-For tables that do not fit in one page enable repeat table titles. {Addressed in all tables}

-Table 1, this is the first time the authors are mentioning a sample size of N=4 872, it is not clear how they arrived at this number. {Has been attended to in the methods through a flow diagram}

-It would be great to stay consistent and use N for total number of participants instead of F as indicated in Table 1. {Has been attended to in all the tables}

-All figures were blurry increasing the resolution of the figures to 600-1000dpi could help solve this problem. {Has been attended to}

-Page 7 “Bivariate analysis of factors associated with medical exclusion among in-migrants and immigrants” please consider rephrasing the paragraph. {Has been attended to}

- For Table 2 please consider labeling the chi2 column P-value and not ꭓ2. {Has been attended to}

-Table 3, it would be great to replace odds label as odds ratio since that is what the authors are presenting. {Has been attended to}

-Table 3 please present one overall p-value for the categorical variables. {Please, the overall p-value for the categorical variables can not be presented as the interpretations for the categorical variables will not be meaningful as comparison with the Reference Category will make it so difficult to do comparisons when carrying out interpretations}

Discussion

-Consider restructuring the discussion and organizing the paragraphs for easy flow of ideas. Please avoid introducing new ideas and concepts. Highlight your findings and substantiate your findings with available literature. {the discussions are restructured and organized for easy flow of ideas}

-Please discuss variables that were statistically significant in the multivariate analysis. Language was not statistically significant.

“Migrants who speak Sesotho, Sepedi and other languages (Other languages are Afrikaans, Swati, Tsonga, Tswana, Venda, Ndebele, Xhosa and English) as their household main languages were more likely to be excluded from health care services compared to those who speak IsiZulu as their household main language” {All findings that are not statistically significant in the multivariate analysis were removed from the discussion as suggested}

Conclusion

-It is clear and linked to the aim.

Ethics

-It is not clear whether this study was cleared by any ethical review committee. {It was submitted upon submission and it was written as follows: Ethics approval and consent to participate: This study only makes use of secondary data without involving any human subjects. Therefore, no formal ethical approval was required. However, the permission to use the data was sought from the GCRO through a written request. Permission was given subject o using the data for this particular research topic only and publishing the findings in a peer-reviewed journal}.

2nd Reviewer’s Responses #2: February 17,2023

Dear Editor,

Thank you for the opportunity to review this manuscript titled: “Health Services Satisfaction and Medical Exclusion among Migrant Youths in Gauteng Province of South Africa: A Cross-sectional Study of the GCRO Survey (2017−2018)”. The study presents an important health inequity issue, using a relatively large sample. However, I have issues with the write-up and results presentation. One major observation is that the background majorly focused on immigrants (foreign nationals) but the sample upon which the analysis was conducted were immigrants (foreign nationals who migrated to the Country) and in-migrants (South African nationals who moved within the Country, a large part of the sample). These different classes of migrants have their peculiar situations and so different concerns or rules will apply to them. Unfavorable policies or health workers’ position towards South Africans, irrespective of whether they migrate to a new Province would differ from that of foreign nationals. Therefore, I have issues with the authors not clearly delineating these differences in their background, and discussion.

See below for detailed comments in relation to different sections of the article:

Abstract

The authors utilized several concepts, many of which were not predefined or explained in the abstract which made it hard to read and follow. The abstract is the first encounter readers will have with an article and if it is very unclear, they may lose interest. [Adjustments has been made in the background of the abstract - Page 1].

- For instance, in the abstract background, they mentioned medical exclusion (the major outcome of the study), and used other terms like medical mistreatment, social phobia and health services satisfaction. I will suggest that the authors avoid shopping around words and should stick to and focus the background on the primary outcome of interest. [These concepts such as medical mistreatment, social phobia, and health services satisfaction are used in literature and several studies to evaluate medical xenophobia. So without these concepts, the primary outcome of interest will be fully explained. However, some adjustments has been made in the background of the abstract - Page 1].

- Also, the study population were migrants which the authors later classified into in-migrants and immigrants, and presented the results using these terms. The authors should consider defining them in the abstract for clarity. [This has been addressed appropriately and adequately in the Abstract background and in the introduction section in Page 1 & Page 2].

Background

Fair.

As earlier mentioned, the background to the study majorly focused on immigrants (foreign nationals), why they immigrate and the impact on their health and the need to access healthcare. While this is clear, the actual study sample involved in-migrants (SA nationals). SA national migrants (in-migrants) who face medical exclusion may be due to completely different issues which were not touched on in the background at all. A lot of the information in the background do not relate to the in-migrants. I suggest you include this important, yet missing information. [This has been addressed appropriately and adequately in Page 2]

Paragraph 2 is quite lengthy, and many sentences were basically repeating the same thing or unnecessary information. Consider shortening this and keeping information that are relevant to the outcome of interest. [This has been addressed appropriately and adequately]

In paragraphs 3 and 4, the authors already highlighted several factors that impact healthcare utilization and access, backed with several citations. Given that, the authors did not provide a proper justification for the current study. The authors stated that there are gaps on how youths are prevented from healthcare services but I don’t think this study address that. Likewise, they stated that they wanted to predictors of healthcare satisfaction and medical exclusion, whereas, the only outcome measured in the study was medical exclusion and satisfaction was merely a predictor. Consider revising your objective or addressing the gap in your result. [This has been addressed appropriately and adequately]

Methods

Well written and detailed enough.

Results

- In the Tables, use n instead of F or write Frequency.[This has been addressed appropriately and adequately in Page 8 & Page 10]

- Place the text describing Table 1 before the Table. [This has been addressed appropriately and adequately in Page 8]

- Consider adding where the result in Figure 2 can be presented. That is, the frequency and percentage of non-immigrants, immigrants and in-migrants. You do not need this information on a separate Figure. [The Figure 2 is removed and the information on Figure 2 was added as Table 1 in the sub-heading of each of the migrant categories by stratification as ‘non-migrant’, ‘in-migrant’ and ‘immigrant’]

- The results descriptions are too detailed. The results are already presented in the Tables, describe the interesting findings, rather than describing all the results in the Table, especially the results for Tables 1 and 3 (extremely long). [This has been addressed appropriately and adequately].

- Rectify the results describing Table 2. Comment in text.

- Some of the frequencies when added up, exceeds the total frequencies for that group. E.g. employment status for the not excluded group was equal to 1626 but you reported the overall sample is 1625. Please make sure all the results presented are correct. [This has been addressed appropriately and adequately]

- The description of results on Table 3 is extremely long and the odds ratios were mostly interpreted incorrectly or unclear. If you have an OR of >1, that is increased odds and OR<1 is decreased odd. Categories with say OR of 0.70 are 30% less likely while OR of say 3.22 will be 3.22 times more likely. Please present interesting findings with clear and accurate interpretations. [This has been addressed appropriately and adequately]

-The use of UAOR (Unadjusted AOR) is incorrect. You either have an adjusted OR or unadjusted OR. AOR means adjusted odd ratio. [This has been addressed appropriately and adequately]

- All the figure are very blurry. Please remake. Consider deleting Figures 1 and 2. Too many information in the manuscript already and Figures seems redundant. [This has been addressed appropriately and adequately]

Discussion, recommendation, implications

- The authors had a very lengthy discussion. The discussion is the “so what” of the study. It should focus on the key take away points, the meaning, the potential reasons underlying them, how they compare with other studies and the implications. [This has been addressed appropriately and adequately]

- The authors did not properly discuss the results for the different groups and the implications. What you would expect among SA national migrants would be different from foreign nationals and proper discussion regarding that should be made. For now, the authors have merged the two classes of migrants (immigrants and in-migrants), who are clearly different and will face different challenges. [This has been addressed appropriately and adequately]

- Aside the lengthy discussion, the authors also had lengthy implications and recommendations. This is a research article and not a thesis. Try to summarize and drive home the key points. [This has been addressed appropriately and adequately]

Conclusion

- The conclusion should be based on the findings. [This has been addressed appropriately and adequately]

- The authors added the prevalence for both groups and indicated the prevalence was 10% and very high. This is in relation to what? [This has been addressed appropriately and adequately]

References

- This is not a review paper. The references are just so many. Delete redundant references.[This has been addressed]

Minor comments

Title: I suggest you change “A cross-sectional study of the GCRO survey” to “A cross-sectional analysis of the GCRO survey” rather than “A cross-sectional study of GCRO survey”. Also, consider spelling out GCRO. [This has been addressed appropriately]

Keywords: Factor is not needed as a keyword [This has been addressed appropriately]

Abstract

- Dissatisfaction of healthcare services should be changed to dissatisfaction with health care services” [This has been addressed appropriately]

Background

- Consider revising this from “Regardless of their gender, they are often victims of violence, infectious diseases, and malnutrition….” to “Regardless of gender, migrants are often victims of violence, infectious diseases, and malnutrition…”. The excessive use of “they” in the paragraph makes it challenging to follow. [This has been addressed appropriately]

- Line 8: The word “ordinary” should be removed “…ordinary non-migrant” as the meaning of ordinary in that context wasn’t clear. [This has been addressed appropriately]

- The latter part of this sentence is unclear “ The trend and pattern of migration ranges from low-income to high-income nations, including those African countries with political, social and economic stability, envisages a growing prevalence of migrant youths”. Consider revising. [This has been addressed appropriately]

- Before ref 15, change migrant people to migrants. [This has been addressed appropriately]

Other minor comments were indicated in the manuscript. 

- would rather thinking that social phobia may serve as an obstacle to healthcare utilization majorly, before considering satisfaction or dissatisfaction. [This has been addressed appropriately]

- Which of the listed categories have the p-values listed? Or are they arranged in order? Rather say all p<0.05. [This has been addressed appropriately and p<0.05 is used]

- Rather say, the adjusted logistic regression showed that only xxxxxxx were independent predictors of medical exclusion. [This has been addressed appropriately]

- Write this in a clearer way [This has been addressed appropriately]

- How realistic is this? Did you mean easy access to healthcare? [This has been addressed appropriately]

- How realistic is this? Did you mean easy access to healthcare? [This has been addressed appropriately]

- Consider revising this to " regardless of gender, migrants are often...." [This has been addressed appropriately]

- The meaning of this word in this context is unclear. Consider removing it. [This has been addressed appropriately]

- This part does not fit well into this sentence. [This has been addressed appropriately]

- change to migrants. [This has been addressed appropriately]

- This sentence does not flow logically with the prior sentence. The prior sentence was pointing out how individual circumstances contribute to poor health, then there was a jump to existing health policies? [This has been addressed appropriately]

- This is a repetition. You already said "stated" at the beginning of the sentence [This has been addressed appropriately]

- This is a general assumption, does it also apply to medical xenophobia? [This is factual and cited authors of such studies are included]

- This is a repetition. You already mentioned gap at the beginning of the sentence. [This has been addressed appropriately]

- This word sounds sentimental. [This has been addressed appropriately]

- There are three different key concepts right here: health service satisfaction, medical exclusion and reasons why migrants are prevented from healthcare services. The prior paragraphs were all over the place and didn't really address these key concepts or how you arrived at them. And also how they are defined for clarity. The prior paragraphs need to be more focused and should set up the ground for these problem statements. [It has been looked at]

- delete this [This has been addressed appropriately]

- How does this relate to the prior sentences? Are you justifying their underrepresentation? [This was justifying their underrepresentation]

- n is a more commonly used sign for sample [This has been addressed appropriately]

- Figure 2 [This has been addressed]

- Just state the number here. 18.7% reported being dissatisfied. [This has been addressed]

- Better tag this as having extra income [This has been addressed]

- Is this a repetition? [No, this is not a repetition]

- what does this represent? Do not assume that the readers know. [This has been addressed]

- Use racial or ethnic group. [This has been addressed and racial group was used. Although, in the datasets, population group was used]

- Reduce these results to the important or interesting variables. The Table is there for more details. [This has been addressed]

- You cannot interpret the whole table. pick the interesting findings. [This has been addressed]

- remove the from the discussion section first line. [This has been addressed]

- higher in relation to what? (Conclusion section). [This has been addressed]

---

## [Decision Letter · Decision Letter 1]

13 Sep 2023

PONE-D-22-33773R1Health Services Satisfaction and Medical Exclusion among Migrant Youths in Gauteng Province of South Africa: A Cross-sectional Analysis of the GCRO Survey (2017−2018)PLOS ONE

Dear Dr. Akokuwebe,

Thank you for submitting your manuscript to PLOS ONE. After careful consideration, we feel that it has merit but does not fully meet PLOS ONE’s publication criteria as it currently stands. Therefore, we invite you to submit a revised version of the manuscript that addresses the points raised during the review process.

We look forward to receiving your revised manuscript.

Kind regards,

Engelbert A. Nonterah, MD, PhD

Academic Editor

PLOS ONE

Journal Requirements:

Additional Editor Comments:

Authors are advised to effect changes in the main manuscript and nt only in the response letter. To this end kindly submit a clean version and a tracked changes version

Reviewers' comments:

Reviewer's Responses to Questions

**Comments to the Author**

1. If the authors have adequately addressed your comments raised in a previous round of review and you feel that this manuscript is now acceptable for publication, you may indicate that here to bypass the “Comments to the Author” section, enter your conflict of interest statement in the “Confidential to Editor” section, and submit your "Accept" recommendation.

Reviewer #3: All comments have been addressed

Reviewer #4: (No Response)

2. Is the manuscript technically sound, and do the data support the conclusions?

Reviewer #3: Yes

Reviewer #4: Yes

3. Has the statistical analysis been performed appropriately and rigorously? 

Reviewer #3: Yes

Reviewer #4: No

4. Have the authors made all data underlying the findings in their manuscript fully available?

Reviewer #3: No

Reviewer #4: No

5. Is the manuscript presented in an intelligible fashion and written in standard English?

Reviewer #3: No

Reviewer #4: Yes

6. Review Comments to the Author

Reviewer #3: Abstract:

The abstract provides a comprehensive overview of the study's key findings and its relevance. However, it's important to include specific quantitative results or effect sizes to give readers a sense of the magnitude of the findings.

Introduction:

In the introduction, consider adding a concise statement of the research objectives or hypotheses. This will help readers understand the specific questions the study aims to address.

Provide a brief rationale for why this study is important. Why is understanding health services satisfaction and medical exclusion among migrant youths in South Africa significant?

Methods:

Include more details on the GCRO survey (e.g., sample size, data collection methods, survey instruments) to help readers understand the data source better.

When discussing statistical analysis, specify the exact statistical tests or modeling techniques used for various analyses. Mention how missing data were handled, if applicable.

Results:

In the results section, consider breaking down the presentation of findings into subsections to make it easier for readers to navigate. For example, you could have subsections for demographics, health facility usage, satisfaction, and exclusion.

When presenting prevalence percentages, consider providing 95% confidence intervals, especially when comparing different groups. This adds precision to the estimates.

For significant findings, briefly discuss their practical implications. What do these results mean for healthcare policies or interventions targeting migrant youths?

Discussion:

The discussion section should go beyond summarizing the results. It's an opportunity to interpret the findings in the context of existing literature. How do the study's results align with or diverge from prior research on this topic?

Consider discussing potential explanations for any unexpected or counterintuitive results. Were there limitations in the study design that could account for these findings?

Highlight the policy implications of the findings. If possible, suggest specific policy changes or interventions that could address the issues identified in the study.

Limitations:

The limitations section is essential. However, it could be more detailed. Mention any potential sources of bias or limitations in the data source (e.g., self-reporting bias, sampling limitations) that could have influenced the results.

Conclusion:

Summarize the key takeaways of the study concisely. Restate the main findings and their implications.

Recommendation:

If there are specific recommendations based on the findings, provide them here. For example, you could recommend changes in healthcare policies or interventions to improve the healthcare experiences of migrant youths.

Acknowledgments:

Ensure that all individuals, institutions, or organizations deserving acknowledgment are included in this section, and consider specifying their contributions if relevant.

Remember to maintain a clear and logical flow between sections and subsections throughout the paper. These comments should help improve the clarity, detail, and overall quality of the paper.

Reviewer #4: 

Authors should include a brief description about how the weighting was implemented for this analysis.

Authors should include more information about the collinearity assessment in the manuscript.

I think that the proportions of those who were medically excluded should be included in the descriptive analysis.

Authors should include some definition of key terms in the introduction “medical exclusion” and others.

The way medical exclusion was conceptualized for this study has some limitation that I was expecting the authors to highlight. If a respondents had personal challenges that prevented him from seeking care, then it does not mean they have been excluded.

Although others provided a link which seem to be a repository for the data used for this analysis it was not included in the main manuscript.

Also, I note that authors used secondary data for this analysis. It will be great for the authors to include ethics approval status for the original survey.

Generally I see that the responses were done directly in the response documents but changes were not effected in the main manuscript.

7. PLOS authors have the option to publish the peer review history of their article (what does this mean?). If published, this will include your full peer review and any attached files.

Reviewer #3: No

Reviewer #4: **Yes: **Solomon Nyame

---

## [Author Response · Author response to Decision Letter 1]

22 Sep 2023

Reviewers’ Comments

1. If the authors have adequately addressed your comments raised in a previous round of review and you feel that this manuscript is now acceptable for publication, you may indicate that here to bypass the “Comments to the Author” section, enter your conflict-of-interest statement in the “Confidential to Editor” section, and submit your "Accept" recommendation.

Reviewer #3: All comments have been addressed

Reviewer #4: (No Response)

 2. Is the manuscript technically sound, and do the data support the conclusions?

Reviewer #3: Yes

Reviewer #4: Yes

 3. Has the statistical analysis been performed appropriately and rigorously? 

Reviewer #3: Yes

Reviewer #4: No

 4. Have the authors made all data underlying the findings in their manuscript fully available?

The PLOS Data policy requires authors to make all data underlying the findings described in their manuscript fully available without restriction, with rare exception (please refer to the Data Availability Statement in the manuscript PDF file). The data should be provided as part of the manuscript or its supporting information, or deposited to a public repository. For example, in addition to summary statistics, the data points behind means, medians and variance measures should be available. If there are restrictions on publicly sharing data—e.g., participant privacy or use of data from a third party—those must be specified.

Reviewer #3: No

Reviewer #4: No

 5. Is the manuscript presented in an intelligible fashion and written in standard English?

Reviewer #3: No

Reviewer #4: Yes

 6. Review Comments to the Author

Reviewer #3: Abstract:

The abstract provides a comprehensive overview of the study's key findings and its relevance. However, it's important to include specific quantitative results or effect sizes to give readers a sense of the magnitude of the findings.

Authors response: 

Thank you very much for your comments. Specific quantitative results/effects sizes of the study were included in the ‘Result Section’ of the Abstract. The specific results included were at the level of the univariate, bivariate and multivariate, as specified in the result section [See Page 0 of the Abstract section in Red colour].

Introduction:

In the introduction, consider adding a concise statement of the research objectives or hypotheses. This will help readers understand the specific questions the study aims to address. 

Authors response: 

The main and specific objective(s) stated below have been inserted in the main manuscript in red colour ink.

Main Objective: 

The primary aim of this study was to document the health services satisfaction and medical exclusion among migrant youths in order to potentially inform future decisions for policy interventions.

Specific objective: 

The specific objectives of this study are to: (1) describe the socio-demographics, economic and health-related characteristics by migration status; (2) to determine the prevalence of medical exclusion and health service satisfaction according to migration status; (3) to examine the factors associated with medical exclusion by migration status; and (4) to examine the predictors of medical exclusion among migrants in Gauteng Province of South Africa. [See Page 3 in Red]

Provide a brief rationale for why this study is important. Why is understanding health services satisfaction and medical exclusion among migrant youths in South Africa significant?

Authors response: The rationale and the significance below has been inserted in the main manuscript in red colour ink.

Using a nationally representative datasets allowed the authors to obtain a representative view on migrants’ perspectives on the satisfaction of health services and; and to investigate the relationships between demographics and medical exclusion practices. Therefore, the rationale for this study is its contribution to an emerging literature that examines health service satisfaction and medical exclusion among migrant youths in South Africa. Hence, its significance, the findings from this study will help to redesign the existing practical interventions that addresses the unmet health needs of youth migrants in South Africa. [See Page 3 in red colour]

Methods:

Include more details on the GCRO survey (e.g., sample size, data collection methods, survey instruments) to help readers understand the data source better.

Authors response:

e.g.,

Sample size - The Fig 1 below shows the diagram illustrating the stages carried out in the sample size selection of in-migrant and immigrant respondents. A two-year cohort of the GCRO study consisted of 24,889 respondents, out of which 4,872 respondents were sampled according to their assigned categories such as non-migrants, in-migrants, and immigrants. A total of 2,162 respondents involving in-migrants and immigrants were then sampled and utilized as the sample size for this study. In this study, the population were migrants aged 18‒29 years, stratified by 1,725 in-migrants and 437 immigrants, totalling 2,162 (See Figure 1). [See Page 5 in red ink colour].

Data collection methods – The GCRO is a nationally representative data in South Africa, and the survey methods employed were included in the main manuscript in red ink colour. However, any information you need to have can be accessed via the GCRO website (https://www.gcro. ac.za) [See Page 4 and Page 5, respectively].

Survey instruments – The survey instruments used were questionnaire, as the GCRO is a quantitative data.

When discussing statistical analysis, specify the exact statistical tests or modeling techniques used for various analyses. 

Authors response: 

The statistical tests of the binary logistic regression, has the dependent variable, which is a dichotomous (binary) variable, coded as 0 or 1. It specifically helps to determine how much a dependent variable (Y) is affected by one or more independent variables (X), where Y is the dependent variable, X is the independent (explanatory) variable, B is the slope and a is the intercept as well as Ɛ is the residual (error). However, the binary regression model is expressed in terms of the logit instead of γ∶logit = Li = βο + βıΧı +⋅⋅⋅+ βκΧκ⋅ [See Page 6 and Page 7, inserted in the main manuscript in red colour]

However,

Univariate analysis explores each variable in a data set, separately. It looks at the range of values, as well as the central tendency of the values. It describes the pattern of response to the variable. It describes each variable on its own. Descriptive statistics describe and summarize data. There are three common methods for performing univariate analysis: (1) Summary Statistics (measures of central tendency and dispersion measures) [Summary statistics information have been included in Table 1 – Page 7 to Page 8], (2) Frequency Distributions, and (3) Charts. [The univariate analysis were used to express these sub-section in the main manuscript such as prevalence of medical exclusion according to migration status, Percentage distribution of the proportion of migrants and the health facility type utilized, and Percentage distribution of migrants and health service satisfaction] [See Page 6 and Page 8 in the main manuscript in red colour ink]

Bivariate analysis is stated to be an analysis of any concurrent relation between two variables or attributes. This study explores the relationship of two variables as well as the depth of this relationship to figure out if there are any discrepancies between two variables and any causes of this difference. The types of bivariate data analysis are the: Numerical and Numerical (in this type, both the variables of bivariate data, independent and dependent, are having numerical values) and Categorical and Categorical (when both the variables are categorical). 

For a bivariate or simple regression with an independent variable x and a dependent variable y, the equation is: y=bx+a, where y is the dependent variable, x is the independent variable, a is the point where the line of best fit intersects the y-axis and b is the angle of the line. [See Page 6 in red colour ink, inserted in the main manuscript in red colour].

Mention how missing data were handled, if applicable.

Authors response: The missing data were found on certain variables where the totals are unequal, so such variables were dropped during the analysis.

Results:

In the results section, consider breaking down the presentation of findings into subsections to make it easier for readers to navigate. For example, you could have subsections for demographics, health facility usage, satisfaction, and exclusion.

Authors response: The presentation findings were broken down into sub-sections for readers to navigate. From Page 7 to Page 9, you will see the sub-sections in bold italics as follows:

-Socio-demographic Characteristics [See Page 7]

-Prevalence of medical exclusion according to migration status [See Page 8]

-Percentage distribution of the proportion of migrants and the health facility type utilized [See Page 8]

-Percentage distribution of migrants and health service satisfaction [See Page 8]

-Bivariate analysis of factors associated with medical exclusion [See Page 8]

-Multivariate analysis of the unadjusted and adjusted logistic regression of the factors associated with medical exclusion [See Page 9]

When presenting prevalence percentages, consider providing 95% confidence intervals, especially when comparing different groups. This adds precision to the estimates.

Authors response: This has been inserted accordingly in Red colour ink.

-Prevalence of medical exclusion according to migration status [See Page 8]

-Percentage distribution of the proportion of migrants and the health facility type utilized [See Page 8]

-Percentage distribution of migrants and health service satisfaction [See Page 8]

For significant findings, briefly discuss their practical implications. What do these results mean for healthcare policies or interventions targeting migrant youths?

Authors response: Practical implications of the study findings for healthcare policies or interventions targeting migrant youths were included in the sub-section: “Implications of findings on medical exclusion of migrants in South Africa”. [See Page 13].

Discussion:

The discussion section should go beyond summarizing the results. It's an opportunity to interpret the findings in the context of existing literature. How do the study's results align with or diverge from prior research on this topic? 

Authors response: The discussion section did not just involve summarizing the results, but it interpreted the findings in the context of existing literature. Summary of results provide the paragraphs and the flow of the findings to align with the context of existing studies. [See Page 11 to Page 13]

Consider discussing potential explanations for any unexpected or counterintuitive results. Were there limitations in the study design that could account for these findings?

Authors response: Thank you very much for your comments. However, discussions on the potential explanation for the results were included in the sub-section: “Strengths and limitations”. [See Page 14]

Highlight the policy implications of the findings. If possible, suggest specific policy changes or interventions that could address the issues identified in the study.

Authors response: Thank you very much for your comments. However, policy implications of the findings were included in the “Implications of findings on medical exclusion of migrants in South Africa”. [See Page 13].

Limitations:

The limitations section is essential. However, it could be more detailed. Mention any potential sources of bias or limitations in the data source (e.g., self-reporting bias, sampling limitations) that could have influenced the results.

Authors response: Limitations of the study is the common methodological limitations of studies. We included all the limitations that are applicable to this study in the sub-section “Strengths and Limitations” [See Page 14].

Conclusion:

Summarize the key takeaways of the study concisely. Restate the main findings and their implications.

Authors response: Thank you very much for your comments. The conclusion is intended to help the reader understand why the research should matter to the readers after they have finished reading the paper, and a conclusion is not merely a summary of points or a re-statement of the research problem but a synthesis of key points. But the significant findings were captured on the conclusion section [See Page 14].

Recommendation:

If there are specific recommendations based on the findings, provide them here. For example, you could recommend changes in healthcare policies or interventions to improve the healthcare experiences of migrant youths.

Authors response: Thank you very much for your comments. However, specific recommendations in this study were grounded on the findings of the study, where recommendations were specifically stated in the “Recommendation section”. [See Page 14].

Acknowledgments:

Ensure that all individuals, institutions, or organizations deserving acknowledgment are included in this section, and consider specifying their contributions if relevant.

Authors response: Thank you very much for your comments. All-important individuals and institutions were full acknowledged in this study [See Page 15].

Remember to maintain a clear and logical flow between sections and subsections throughout the paper. These comments should help improve the clarity, detail, and overall quality of the paper.

Authors response: Thank you very much for your comments. This manuscript was sent to a Professional Editor to edit it for a clear and logical flow throughout the paper between sections and sub-sections.

Reviewer #4: 

Authors should include a brief description about how the weighting was implemented for this analysis.

Authors response: A brief description of weighting was inserted in the main manuscript. [See Page 7 in Red colour ink].

Authors should include more information about the collinearity assessment in the manuscript.

Authors response: Multicollinearity assessment was carried out with STATA statistical software. It is documented in the main manuscript [See Page 7 in Red colour ink].

I think that the proportions of those who were medically excluded should be included in the descriptive analysis.

Authors response: The proportions of those who were medically excluded have been included in the descriptive analysis (Table 2) [See Page 9 to Page 10 in Red colour ink].

Authors should include some definition of key terms in the introduction “medical exclusion” and others.

Authors response: Thank you very much for your comment. Medical exclusion was used in this study and the definition was mentioned in Page 5 of the manuscript. However, we only stated that some studies used the one word ‘medical xenophobia’ as our study used medical exclusion, and we defined what we meant as medical exclusion in this study. [Page 5]

The way medical exclusion was conceptualized for this study has some limitation that I was expecting the authors to highlight. If a respondents had personal challenges that prevented him from seeking care, then it does not mean they have been excluded.

Authors response: The definition of medical exclusion used in this study was clearly defined in the sub-section “Variable Measurement” [See Page 5 in Red colour ink].

Although others provided a link which seem to be a repository for the data used for this analysis it was not included in the main manuscript.

Authors response: Thank you for the comment, but as a result of ethical concerns the data cannot be shred or kept in the repository. We need to comply with the ethics framework not to share the data, however, we deposited the Do-files we created for the analysis as Appendices. [See Page 5 in Red colour ink].

Also, I note that authors used secondary data for this analysis. It will be great for the authors to include ethics approval status for the original survey.

Authors response: Thank you for the comment, we have included the ethics approval and consent to participate in the main manuscript. [See Page 7].

Generally, I see that the responses were done directly in the response documents but changes were not effected in the main manuscript.

Authors response: See all changes made in red colour ink in the main manuscript. [See Page 0 to Page 19].

---

## [Editor Report · Decision Letter 2]

24 Oct 2023

Health Services Satisfaction and Medical Exclusion among Migrant Youths in Gauteng Province of South Africa: A Cross-sectional Analysis of the GCRO Survey (2017−2018)

PONE-D-22-33773R2

Dear Dr. Monica Ewomazino Akokuwebe,

We’re pleased to inform you that your manuscript has been judged scientifically suitable for publication and will be formally accepted for publication once it meets all outstanding technical requirements.

Kind regards,

Engelbert A. Nonterah, MD, PhD

Academic Editor

PLOS ONE
---

## [Editor Report · Acceptance letter]

14 Nov 2023

PONE-D-22-33773R2 

Health Services Satisfaction and Medical Exclusion among Migrant Youths in Gauteng Province of South Africa: A Cross-sectional Analysis of the GCRO Survey (2017−2018) 

Dear Dr. Akokuwebe:

I'm pleased to inform you that your manuscript has been deemed suitable for publication in PLOS ONE. Congratulations! Your manuscript is now with our production department. 

Kind regards, 

on behalf of

Dr. Engelbert Adamwaba Nonterah 

Academic Editor

PLOS ONE